# Investigating Action Encodings in Recurrent Neural Networks in Reinforcement Learning

**Matthew Schlegel**                                           *mkschleg@ualberta.ca*
*University of Alberta*

**Volodymyr Tkachuk**                                          *vtkachuk@ualberta.ca*
*University of Alberta*

**Adam White**                                                      *amw8@ualberta.ca*
*University of Alberta*

**Martha White**                                                 *whitem@ualberta.ca*
*University of Alberta*

**Reviewed on OpenReview:** *https: // openreview. net/ forum? id=K6g4MbAC1r*

## Abstract

Building and maintaining state to learn policies and value functions is critical for deploying reinforcement learning (RL) agents in the real world. Recurrent neural networks (RNNs) have become a key point of interest for the state-building problem, and several large-scale reinforcement learning agents incorporate recurrent networks. While RNNs have become a mainstay in many RL applications, many key design choices and implementation details responsible for performance improvements are often not reported. In this work, we discuss one axis on which RNN architectures can be (and have been) modified for use in RL. Specifically, we look at how action information can be incorporated into the state update function of a recurrent cell. We discuss several choices in using action information and empirically evaluate the resulting architectures on a set of illustrative domains. Finally, we discuss future work in developing recurrent cells and discuss challenges specific to the RL setting.

## 1 Introduction

Learning to behave and predict using partial information about the world is critical for applying reinforcement learning (RL) algorithms to large complex domains. For example, a deployed automated spacecraft with a faulty sensor that is only able to read signals intermittently. For the spacecraft to stay in service it needs to deploy a learning algorithm to maintain helpful information (or state) about the history of intermittent sensor readings as it relates to the other sensors and how the spacecraft is behaving. A game playing system such as StarCraft (Vinyals et al., 2019) provides another good example. An agent who plays StarCraft must build a working representation of the map, it's base and strategy, and any information about its rival's base and strategy as it focuses its observations on specific locations to perform actions.

Deep reinforcement learning has expanded the types of problems reinforcement learning can be applied to, specifically those with complex observations from the environment (Mnih et al., 2015; Vinyals et al., 2019). Significant work has gone into engineering non-recurrent networks (Hessel et al., 2018; Espeholt et al., 2018),

while several challenges remain for recurrent architectures in reinforcement learning (Hausknecht and Stone, 2015; Zhu et al., 2017; Igl et al., 2018; Rafiee et al., 2022; Schlegel et al., 2021). There are many design and algorithmic decisions required when applying a recurrent architecture to a reinforcement learning problem. We have a more detailed discussion on the open-problems for recurrent agents in Section 6.

Recurrent neural networks (RNNs) have been established as an important tool for modeling data with temporal dependencies. They have been primarily used in language and video prediction (Mikolov et al., 2010; Wang and Cho, 2016; Saon et al., 2017; Wang et al., 2018; Oh et al., 2015), but have also been used in traditional time-series forecasting (Bianchi et al., 2017) and RL (Onat et al., 1998; Bakker, 2002; Wierstra et al., 2007; Hausknecht and Stone, 2015; Heess et al., 2015). Many specialized architectures have been developed to improve learning with recurrence. These architectures are designed to better model long temporal dependence and avoid saturation including, Long-short Term Memory units (LSTMs) (Hochreiter and Urgen Schmidhuber, 1997), Gated Recurrent Units (GRUs) (Cho et al., 2014; Chung et al., 2014), Non-saturating Recurrent Units (NRUs) (Chandar et al., 2019), and others. Most modern RNN architectures integrate information through additive operations. However, some work has also examined multiplicative updating (Sutskever et al., 2011; Wu et al., 2016) which follows from what were known as Second-order RNNs (Goudreau et al., 1994).

One important design decision is the strategy used to incorporate action in the state update function which can have a large impact on the agent's ability to predict and control (see Figure 1). This has been noted before, Zhu et al. (2017) provides a discussion on the importance of these choices developing an architecture which encodes the action through several layers before concatenating with the observation encoding. Other types of action encodings have been used for the state update in RNNs for RL (Schaefer et al., 2007; Zhu et al., 2017; Schlegel et al., 2021), but without an in-depth discussion or focus on the ramifications of the particular choice of architecture. In other cases, action has seemingly been omitted (Oh et al., 2015; Hausknecht and Stone, 2015; Espeholt et al., 2018). Other state construction approaches also see action as a primary component, predictive representations of state encode predictions as the likelihood of seeing action-observation pairs given a history (Littman and Sutton, 2002).

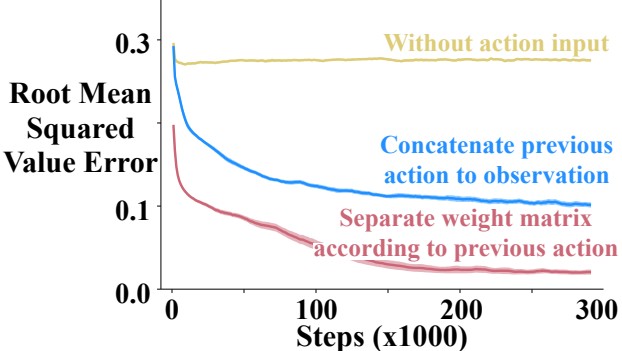

Figure 1: Learning Curves for various RNN cells in Ring World using experience replay and three strategies to incorporate action into an RNN. The agent learns 20 GVF predictions for 300k steps and we report root mean squared value error averaged over 50 runs with 95% confidence intervals with window averaging over 1000 steps. See Section 5.1 for full details.

Action plays an important role in perception in cognitive sciences. Noë (2004) proposed that perception is dependent on the actions we can take and have taken on the world around us. In effect, one can look at the objective of a reinforcement learning agent as the desire to control and predict the experience (or data) stream, which inevitably means we must model our agency on the data stream. Action has also played an important part in understanding representations (or codings) in the brain through common coding (Prinz, 1990), and in the larger interplay between prediction and action in the brain (Clark, 2013). While the RNN architecture is not exactly reminiscent of these cognitive models, the role of action in perception further motivates the need to study the role action plays in an RL agent's perceptual system more in-depth.

In this paper, we focus on several architectures for incorporating action into the state-update function of an RNN in partially observable RL settings. Many of these architectures have been proposed previously for recurrent architectures (i.e. Zhu et al. (2017); Schlegel et al. (2021)), and others are either related to or obvious extensions of those architectures. We perform an in-depth empirical evaluation on several illustrative domains, and outline the relationship between the domain and architectures. Finally, we discuss future work in developing recurrent architectures designed for the RL problem and discuss challenges specific to the RL setting needing investigation in the future.

## 2 Problem Setting

We formalize the agent-environment interaction as a partially observable markov decision processes (POMDP). The underlying dynamics are defined by a tuple $(\mathcal{S}, \mathcal{A}, \mathbf{P}, f_{\mathbf{o}}, \mathcal{R})$. Given a state $\psi \in \mathcal{S}$ and $a \in \mathcal{A}$ the environment transitions to a new state $\psi\prime \in \mathcal{S}$ according to the state transition probability matrix $\mathbf{P} : \mathcal{S} \times \mathcal{A} \times \mathcal{S} \to [0, \infty)$ with a reward given by $\mathcal{R} : \mathcal{S} \times \mathcal{A} \to \mathbb{R}$. The agent observes the sequence $\mathbf{o}_t, a_t, r_{t+1}, \mathbf{o}_{t+1}, a_{t+1}, \dots$ where the observations are a lossy function over the state $\mathbf{o}_t \stackrel{\text{def}}{=} f_{\mathbf{o}}(\psi_t) \in \mathbb{R}^m$, the actions are selected by the agent's current policy $a_t \sim \pi(\cdot | \mathbf{o}_0, a_0, \dots, a_{t-1}, \mathbf{o}_t) \to [0, \infty)$, and the reward is $r_t \stackrel{\text{def}}{=} f_r(\psi_0, \psi_1, \dots, \psi_t) \in \mathbb{R}$.

In this paper we perform experiments in two settings: prediction and control. For prediction, general value functions (GVFs) define the targets (Sutton et al., 2011; White, 2015). A GVF is a tuple containing a cumulant $c_{t+1} = f_c(o_t, a_t, o_{t+1}, r_{t+1}) \in \mathbb{R}$, a continuation function $\gamma_{t+1} = f_\gamma(o_t, a_t, o_{t+1}) \in [0, 1]$, and a history $\mathbf{h}_t = [a_0, \mathbf{o}_1, a_1, \mathbf{o}_2, a_2, \dots, \mathbf{o}_t]$ conditioned policy $\pi(a_t | \mathbf{h}_t) \in [0, \infty)$. The goal of the agent is to learn a value function which estimates the expected cumulative return under $\pi$,

$$\mathbb{E}_\pi\left[G_t^c | H_t = \mathbf{h}_t\right] \qquad \text{where } G_t^c \stackrel{\text{def}}{=} c_{t+1} + \gamma_{t+1} G_{t+1}^c.$$

To estimate the value function we use off-policy semi-gradient TD(0) (Sutton, 1988; Tesauro et al., 1995). For the control setting we learn a policy which maximizes the discounted sum of rewards or return $G_t \stackrel{\text{def}}{=} \sum_{i=0}^\infty \gamma^i r_{i+t+1}$. In this paper, we use Q-learning (Watkins and Dayan, 1992) to construct an action-value function and take actions according to an epsilon-greedy strategy.

## 3 Constructing State with Recurrent Networks

For effective prediction and control, the agent requires a state representation $\mathbf{s}_t \in \mathbb{R}^n$ that is a sufficient statistic of the past: $\mathbb{E}[G_t^c | \mathbf{s}_t] = \mathbb{E}[G_t^c | \mathbf{s}_t, \mathbf{h}_t]$. When the agent learns such a state, it can build policies and value functions without the need to store any history. For example, for prediction, it can learn $V(\mathbf{s}_t) \approx \mathbb{E}[G_t^c | \mathbf{s}_t]$. In this section, we describe the strategies used in this work to learn state.

An RNN provides one such solution to learning $\mathbf{s}_t$ and associated state update function. The simplest RNN is one which learns the parameters $\boldsymbol{\theta} \in \mathbb{R}^d$ recursively

$$\mathbf{s}_t = \sigma(\boldsymbol{\theta} \mathbf{x}_t + \mathbf{b})$$

where $\mathbf{x}_t = [\mathbf{o}_t, \mathbf{s}_{t-1}]$ and $\sigma$ is any non-linear transfer function (typically tanh). While concatenating information (or doing additive operations) has become standard in RNNs, multiplicative operations have also been explored

$$(\mathbf{s}_t)_i = \sigma\left(\sum_{j=1}^M \sum_{k=1}^N \boldsymbol{\theta}_{ijk}(\mathbf{o}_t)_j(\mathbf{s}_{t-1})_k + \mathbf{b}_i\right) \qquad \triangleright \text{ where } \boldsymbol{\theta} \in \mathbb{R}^{|\mathbf{s}| \times |\mathbf{o}| \times |\mathbf{s}|}.$$

Using this type of operation was initially called second-order RNNs (Goudreau et al., 1994), and was explored in a landmark success of RNNs (Sutskever et al., 2011) in a character-level language modeling task.

RNNs are typically trained through the use of back-propagation through time (BPTT) (Mozer, 1995). This algorithm effectively unrolls the network through the sequence and calculates the gradient as if it was one large network with shared weights. This unrolling is often truncated at some number of steps $\tau$. While this alleviates computational-cost concerns, the learning performance can be sensitive to the truncation parameter (Pascanu et al., 2013). When calculating the gradients through time for a specific sample, we follow (Schlegel et al., 2021) and define our loss as

$$\mathcal{L}_t(\boldsymbol{\theta}) = \sum_i^N (v_i(\mathbf{s}_t(\boldsymbol{\theta})) - y_{t,i})^2$$

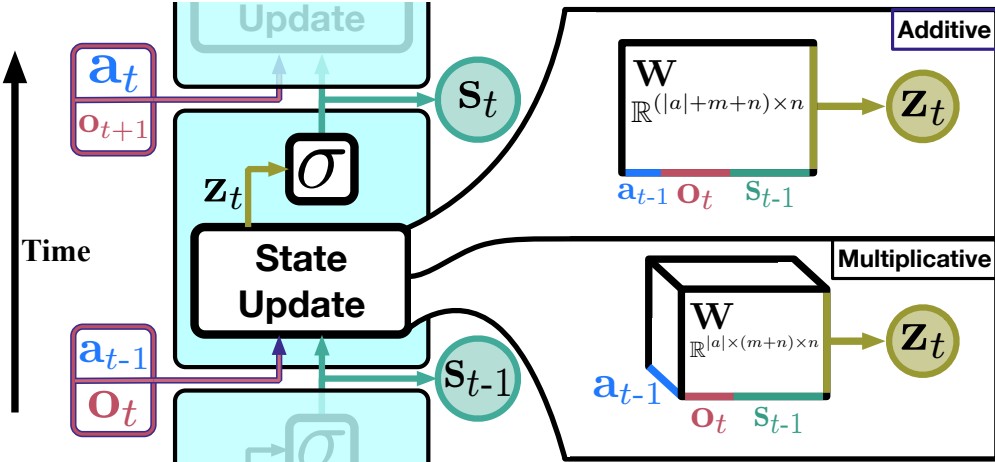

Figure 2: Visualizations of the multiplicative and additive RNNs. The dimensions of the weight matrices use the size of the RNN's state $|s_{t-1}| = n$ and the size of the observation $|o_t| = m$.

where $N$ is the size of the batch, and $y$ is the target defined by the specific algorithm. This effectively means we are calculating the loss for a single step and calculating the gradients from that step only.

There are several known problems with simple recurrent units (and to a lesser extent other recurrent cells). The first is known as the vanishing and exploding gradient problem (Pascanu et al., 2013). In this, as gradients are multiplied together (via the chain rule in BPTT) the gradient can either become very large or vanish into nothing. In either case, the learned networks often cannot perform well and a number of practical tricks are applied to stabilize learning (Bengio et al., 2013). The second problem is called saturation. This occurs when the weights $\boldsymbol{\theta}$ become large and the activations of the hidden units are at the extremes of the transfer function. While not problematic for learning stability, this can limit the capacity of the network and make tracking changes in the environment dynamics more difficult (Chandar et al., 2019). Because of these issues, several variations on the simple recurrent cell have been developed including the LSTMs, GRUs, and NSRUs. We focus our experiments around the simple recurrent cells (RNNs) and GRUs.

Finally, to improve sample efficiency we incorporate experience replay (ER), a critical part of a deep (recurrent) system in RL (Mnih et al., 2015; Hausknecht and Stone, 2015). There are two key choices here: how states are stored and updated in the buffer and how sequences are sampled (Kapturowski et al., 2018). We store the hidden state of the cell in the experience replay buffer as apart of the experience tuple. This is then used to initialize the state when we sample from the buffer for both the target and non-target networks. We pass back gradients to the stored state to update them along with our model parameters, see a full discussion in Section 6. We also stored a separate initial state for the beginning of episodes, which was updated with gradients. We slightly differ from the approach taken by Kapturowski et al. (2018), but expect this architectural choice to have little impact on our discussion in this paper. If we sampled the beginning of an episode from the replay we used the most up to date version of this vector to initialize the hidden state. For sampling, we allowed the agent to sample states across the episode. For samples at the end of the episode, we simply use a shorter sequence length than $\tau$.

## 4 Architectural Designs for Incorporating Action

In this paper, we define two broad categories for incorporating action into the state update function of an RNN, and discuss various variations on these ideas (see Figure 2 for a visualization of two main architectures).

### 4.1 Additive

The first category is to use an additive operation. The core concept of additive action recurrent networks is concatenating an action embedding as an input into the recurrent cell (Schaefer et al., 2007; Zhu et al.,

2017). For example, the update becomes

$$\mathbf{s}_t = \sigma\left(\mathbf{W}^{\mathbf{x}}\mathbf{x}_t + \mathbf{W}^{\mathbf{a}}\mathbf{a}_{t-1} + \mathbf{b}\right) \qquad \textbf{(Additive)}$$

where $\mathbf{W}^{\mathbf{x}}$ and $\mathbf{W}^{\mathbf{a}}$ are appropriately sized weight matrices. This requires no changes to the recurrent cell if the action embedding $\mathbf{a}_{t-1} \in \mathbb{R}^b$ if concatenated to the observation vector. In the empirical experiments, the additive update cells use a hand-designed one-hot encoding function as all our domains have discrete actions.

A variant of the additive approach was explored in Zhu et al. (2017), where they modified the architecture slightly to learn a function of the action input $\mathbf{a}_t = f_a(a_t)$. In this paper, we use the label **Deep Additive** for the architecture, where the action encoding function $f_a$ is a feed-forward neural network. As in their architecture, we concatenate the action embedding with the observation encodings right before the recurrent network. This enables us to focus on the changes in the basic operation rather than enumerating all possible places the action can be concatenated before the recurrent operation.

### 4.2 Multiplicative

The second category is inspired by second-order RNNs (Goudreau et al., 1994) and first appeared as a part of a state update function in Rafols et al. (2006), where the observation, hidden state, and action embedding are integrated using a multiplicative operation:

$$\mathbf{s}_t = \sigma\left(\mathbf{W} \times_2 \mathbf{x}_t \times_3 \mathbf{a}_{t-1}\right), \qquad \textbf{(Multiplicative)}$$

where $\mathbf{W} \in \mathbb{R}^{|\mathbf{s}_t| \times |\mathbf{x}_t| \times |\mathbf{a}_{t-1}|}$ and $\times_n$ is the $n$-mode product, which we detail in Appendix D. This type of operation is known to expand the types of functions learnable by a single layer RNN (Goudreau et al., 1994; Sutskever et al., 2011), and decreases the networks sensitivity to truncation (Schlegel et al., 2021).

While this type of update has very clear advantages, there is also a tradeoff in terms of number of parameters and potential re-learning depending on the granularity of the action representation. For example, in the Ring World experiment above the RNN cell with additive used 285 parameters with hidden state size of 15. The multiplicative version would have used 510 parameters with the same hidden state size. While this doesn't seem like a lot, if we compare what it would be in a domain like Atari (with 18 actions, 1024 inputs, and $|s_t| = 1024$) the number of parameters would be ~2 million vs ~38 million respectively. As shown below in the empirical study, the size of the state can be significantly reduced when using a multiplicative update. In any case, it would be worthwhile to develop strategies to reduce the number of parameters, which we discuss next.

### 4.3 Reducing parameters of the Multiplicative

The first way we can reduce the number of parameters is by using a low-rank approximation of the tensor operations. Like matrices, tensors have a number of decompositions which can prove useful. For example, every tensor can be factorized using canonical polyadic decomposition, which decomposes an order-N tensor $\mathbf{W} \in \mathbb{R}^{I_1 \times I_2 \times \ldots \times I_N}$ into n matrices as follows

$$\mathbf{W}_{i_1, i_2, \ldots} = \sum_{r=1}^{M} \lambda_r \mathbf{W}^{(1)}_{i_1, r} \mathbf{W}^{(2)}_{i_2, r} \ldots \mathbf{W}^{(N)}_{i_N, r}$$

where $\mathbf{W}^{(j)} \in \mathbb{R}^{I_j \times M}$, $\lambda_r \in \mathbb{R}$ is the weighting for factor $r$, and $M$ is the rank of the tensor. This is a generalization of matrix rank decomposition and exists for all tensors with finite dimensions, see Appendix D for more details. We can make several simplifications using the properties of n-mode products. Using the definition of the multiplicative RNN update,

$$\mathbf{W} \times_2 \mathbf{x}_t \times_3 \mathbf{a}_{t-1} \approx \boldsymbol{\lambda}\mathbf{W}^{out}\left(\mathbf{x}_t\mathbf{W}^{in} \odot \mathbf{a}_{t-1}\mathbf{W}^a\right)^{\top} \quad \triangleright \boldsymbol{\lambda}_{i,i} = \lambda_i. \qquad \textbf{(Factored)}$$

Previous work explored using a low-rank approximation of a multiplicative operation. A multiplicative update was used to make action-conditional video predictions in Atari (Oh et al., 2015). This operation also

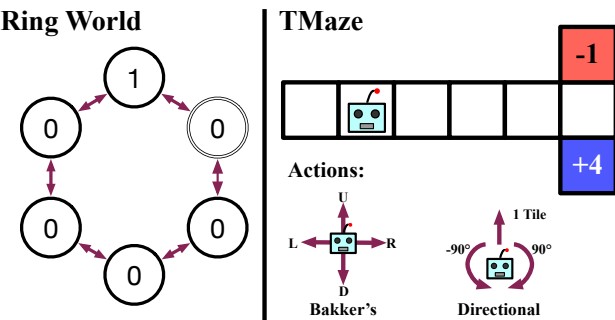

Figure 3: The illustrative environments used in Section 5.1 and Section 5.2 respectively. (**left**) The Ring World environment with 6 states is depicted, where the observation the agent receives is denoted in each of the circles, available actions denoted by the red arrows, and the agent's current location denoted by a double line. (**right**) The base TMaze environments are depicted with the available actions denoted below and labeled according to the Bakker's TMaze and Directional TMaze used in Section 5.2.

appears in a Predictive State RNN hidden state update (Downey et al., 2017), albeit it never performed as well as the full rank version. Our low rank approximation is also similar to the network used in Sutskever et al. (2011), where they mention optimization issues (which were overcome through the use of quasi-second order methods).

Another approach to reducing the number of parameters required—and to reduce redundant learning—by using an action embedding rather than a one-hot encoding. For example, in Pong it is known that only 5 actions matter. By taking advantage of the structure of the action space we could potentially further reduce the number of parameters required to get these benefits. We explore this architecture briefly in Section E.5. While this is an important piece of the puzzle, we do not focus on learning good action embeddings in this paper and leave it to future work.

## 5 Experiments

In the following sections, we set out to empirically evaluate the three operations for incorporating action into the state update function: **N**o **A**ction input ("**NA**"), **A**dditive **A**ction ("**AA**"), **M**ultiplicative **A**ction ("**MA**"), **Fac**tored ("**Fac**"), **D**eep **A**dditive **A**ction ("**DAA**"). We explore all the variants using both standard RNNs and a GRU cell. Our experiments are primarily driven by the main hypothesis that the multiplicative will strictly outperform the other variants, as suggested by Schlegel et al. (2021). To explore this hypothesis we focus on two main empirical questions:

1. How do the different cells affect the properties of the learned value function and internal state of the agent?

2. Are there examples where the other variants outperform the multiplicative variant?

**Question 1:**

There are several properties we are interested in when analyzing the learning capabilities of our agent. First, and most obvious, is prediction error (calculated using root mean squared value error). While error is a reasonable method to compare different architectures, Kearney et al. (2020) argue only inspecting error can be misleading in the quality of the prediction. To account for this in our analysis we visually inspect the raw predictions as well to confirm they are reasonably modeling the target returns. With respect to the internal state, we are primarily interested in understanding if there are qualitative differences which lead to differences in prediction quality.

**Question 2:**

The second question is more straightforward than the first, and requires a complete empirical investigation of all the variants on a set of problems with a diverse set of underlying dynamics and characteristics. You can see this question as an extension of the hypothesis implied by Figure 1 and Schlegel et al. (2021):

> The multiplicative update outperforms the other variants in the reinforcement learning setting for both control and prediction.

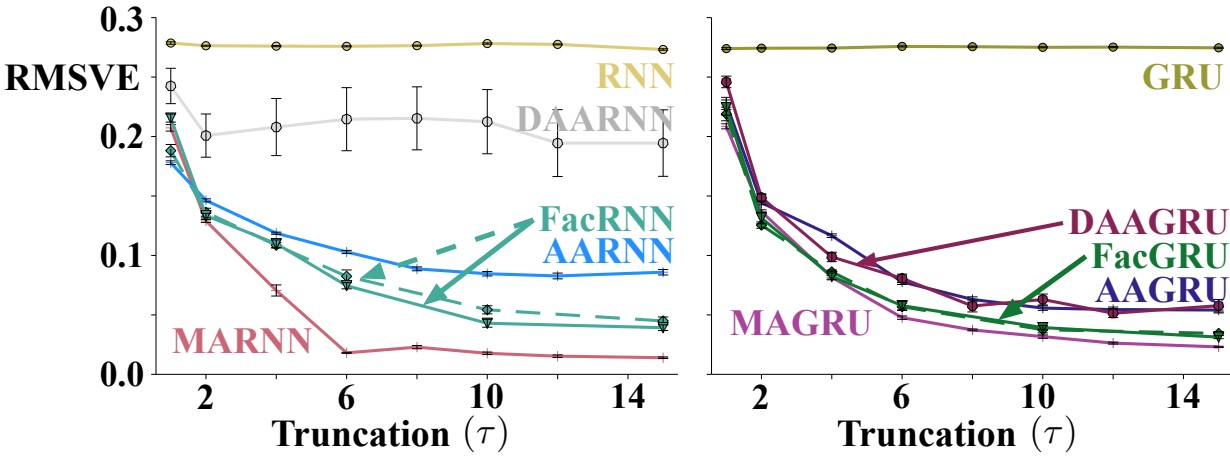

Figure 4: Ring World sensitivity curves of RMSVE over the final 50k steps for CELL (hidden size) **(left)** RNN (15), AARNN (15), MARNN (12), FacRNN (12 [solid] and 15 [dashed]), DARNN (12, $|\mathbf{a}| = 2$), and **(right)** GRU (12), AAGRU (12), MAGRU (9), FacGRU (9 [solid] and 12 [dashed]), DAGRU (9, $|\mathbf{a}| = 10$). Reported results are averaged over 50 runs with a 95% confidence interval. FacRNN used factors $M = \{12, 8\}$ respectively, and FacGRU used $M = \{14, 12\}$. All agents were trained over 300k steps.

While we cannot confirm the above hypothesis empirically, if question 2 is affirmed the hypothesis is false. Counter examples for the hypothesis will also lead to more intuitive knowledge about when to apply one of the above variants.

**Other details:**

In all control experiments, we use an $\epsilon$-greedy policy with $\epsilon = 0.1$. All networks are initialized using a uniform Xavier strategy (Glorot and Bengio, 2010), with the multiplicative operation independently normalizing across the action dimension (i.e. each matrix associated with an action in the tensor is independently sampled using the Xavier distribution). Unless otherwise stated, we performed a hyperparameter search for all models using a grid search over various parameters (listed appropriately in the Appendix F). To best to our ability we kept the number of hyperparameter settings to be equivalent across all models, except the factored variants which use several combinations of hidden state size and number of factors. The best settings were selected and reported using independent runs with seeds different from those used in the hyperparameter search, unless otherwise specified. We controlled all the network sizes such that they had an approximately equal number of free parameters. All final network sizes can be found in Appendix F.

All experiments were run using an off-site cluster. In total, for all sweeps and final experiments we used $\sim 20$ cpu years, which was approximated based off the logging information used by the off-site cluster. All code for the following experiments can be found at `https://github.com/mkschleg/ActionRNNs.jl` and is written in Julia (Bezanson et al., 2017), and we use Flux and Zygote as our deep learning and auto-diff backend (Innes, 2018b;a).

### 5.1 Investigating Properties of Predictions and State

We explore the first empirical qeustion by revisiting the Ring World environment, specifically to test model performance with various truncations, and to compare the architecture's learned state. The Ring World, depicted in Figure 3, consists of a cycle of states with a single state containing an active observation bit, and other states having an inactive observation bit. The agent can take actions moving either clockwise or counter clockwise in the cycle of states. The agent must keep track of how far it has moved from the active bit. For all experiments we use a Ring World with 10 underlying states.

The agent's objective is to learn a total of 20 GVFs with state-termination continuation functions of $\gamma \in \{0.0, 0.1, 0.2, 0.3, 0.4, 0.5, 0.6, 0.7, 0.8, 0.9\}$. When the agent observes the active bit in Ring World (i.e. enters the first state) the predictions are terminated (i.e. $\gamma = 0.0$). The GVFs use the observed bit as a cumulant.

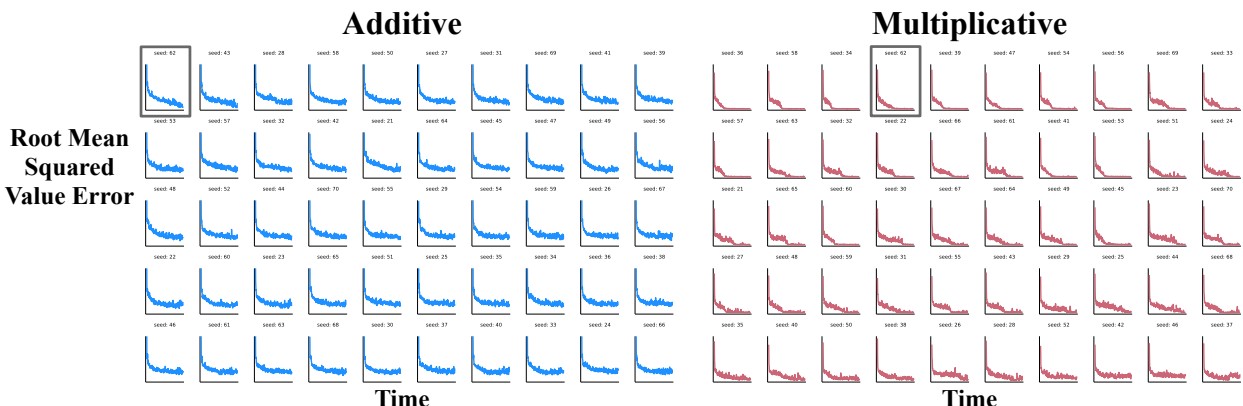

Figure 6: Individual learning curves for the additive (hidden size of 15) and multiplicative (hidden size 12) RNNs in Ring World with truncation $\tau = 6$. The plots are smoothed with a moving average with 1000 step window sizes. The gray box denotes the seed used in Figures 5 and 7. Overall, we see the multiplicative is quite resilient to initialization, but the distance from zero error in Figure 1 can be explained by a few bad initializations.

Half follow a persistent policy of going clockwise and the other follow the opposite direction persistently. The agent follows an equiprobable random behavior policy. The agent updates its weights on every step following a off-policy semi-gradient TD update with a truncation values denoted. We train the agent for 300000 steps and averaged over 50 independent runs. We use root mean squared value error (RMSVE) as the core error metric, which is $\text{RMSVE}_t = \frac{1}{|V(h_t)|}||V(h_t) - V_{\text{oracle}}(\psi_t)||_2$, where $V_{\text{oracle}}$ is a known oracle for the true value function.

**Results:**

We start with a survey over truncation values for all the architectures in Figure 4. For both the RNN and GRU cells the MA variant performs the best, while the additive performs the worst of the cells which include action information. Interestingly, the factored variants for the GRU perform almost identically, while the FacRNN with a smaller hidden state perform marginally better. All factored variants straddled the performance of the additive and multiplicative updates. The DAAGRU performs similarly to the AAGRU, while the DAARNN fails to learn in this setting. Finally, the MARNN performs the best overall, only needing a truncation value of $\tau = 6$ to learn, which is shorter than the Ring World. We conclude that with the same number of parameters, the operation used to update the state can have a significant effect on the required sequence length and final performance.

To ground the prediction error reported, we present two representative examples of the learned predictions for the additive and multiplicative RNNs in Figure 5. These plots show a single seed (selected as the best for the additive) over a small snippet of time, but are representative of our observations of the general performance for both cells. The multiplicative follows the actual prediction within a small delta being as close to zero error as we should expect, while the additive has many artifacts and other miss-predictions for both the myopic ($\gamma = 0.0$) and long-horizon ($\gamma = 0.9$) predictions. In Figure 6, we report all the individual learning curves for the additive and multiplicative.

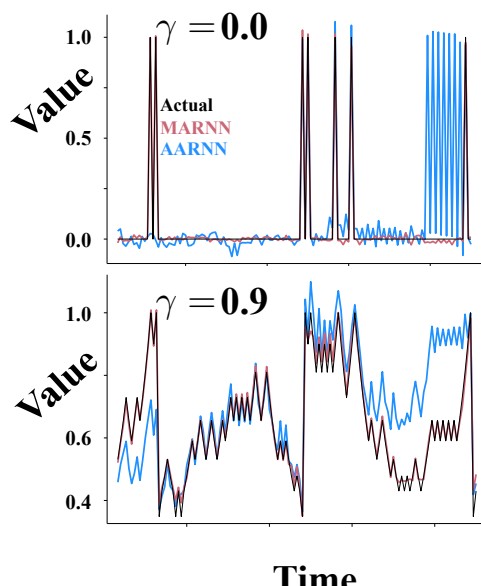

Figure 5: Ring World predictions of seed = 62 for the multiplicative and additive RNNs. Discounts listed with the target policy persistently going counter-clockwise.

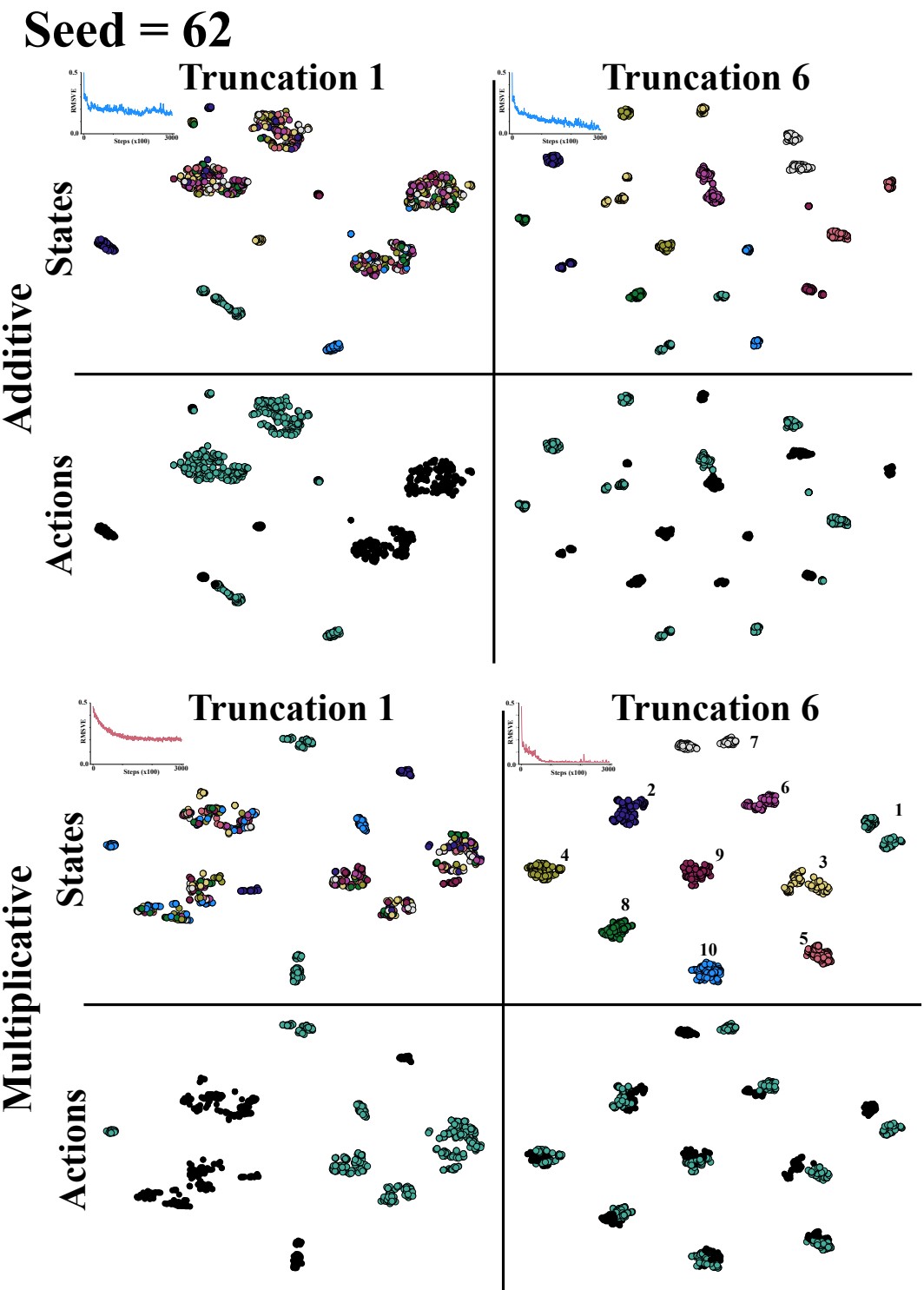

Figure 7: TSNE plots for the additive and multiplicative RNNs for truncation $\in \{1, 6\}$. Given the learning objective (described in Section 5.1), we would want the state to have 10 distinct clusters for each state of the underlying environment. We should expect the truncation $\tau = 1$ to not be able to produce this kind of state for either cell variant. The learning curves correspond to a single seed (seed=62 which is best for the Additive update). The top scatter plots are colored on the underlying state the agent is currently in, the bottom scatter plots are colored based on the previous action the agent took. We initialized TSNE with the same random seed, with max iterations set to 1000, and perplexity set to 30. We present **(top)** additive and **(bottom)** multiplicative update functions.

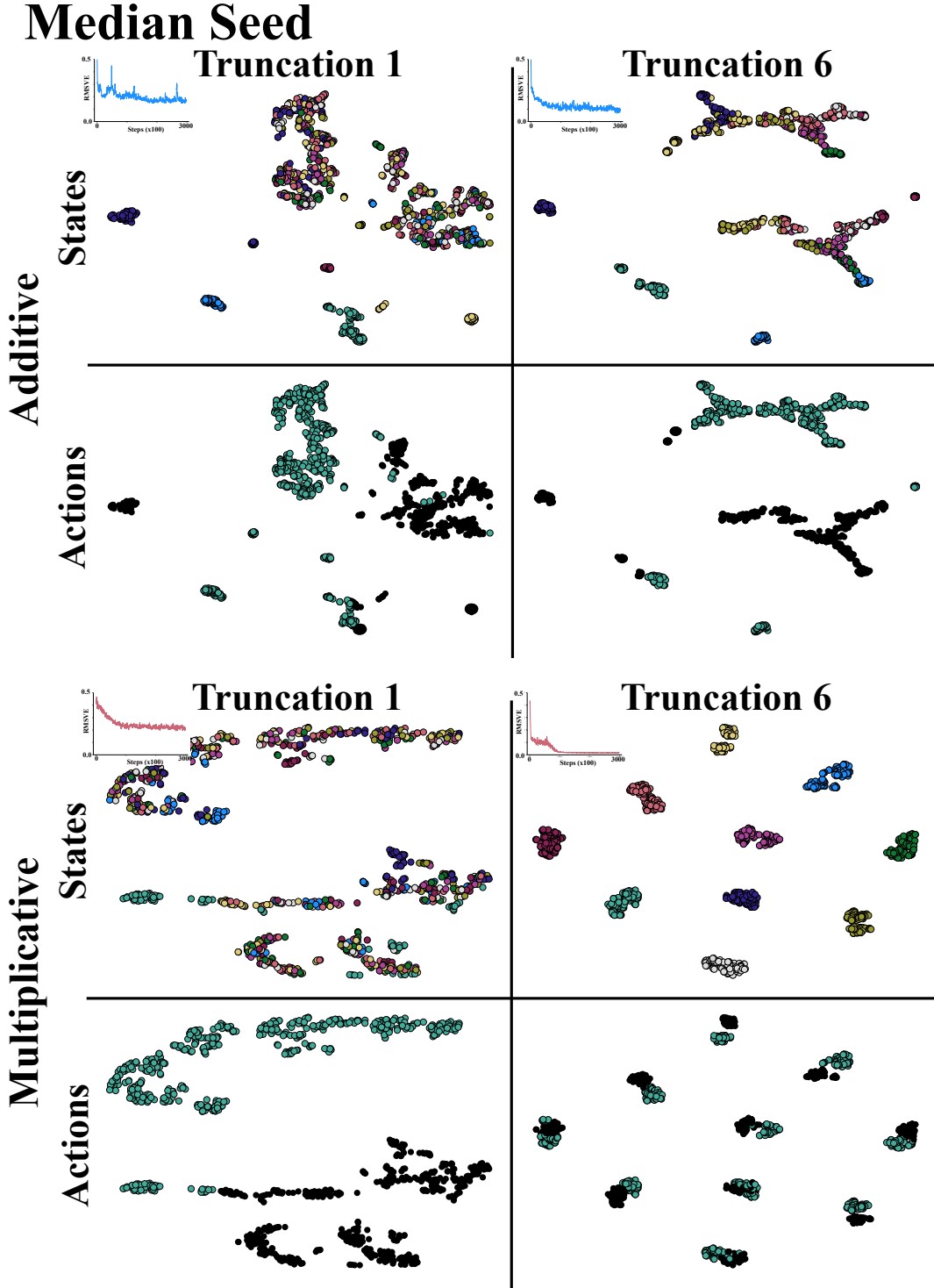

Figure 8: TSNE plots for the additive and multiplicative RNNs for truncation $\in \{1, 6\}$. Given the learning objective (described in Section 5.1), we would want the state to have 10 distinct clusters for each state of the underlying environment. We should expect the truncation $\tau = 1$ to not be able to produce this kind of state for either cell variant. The learning curves correspond to a single seed. The top scatter plots are colored on the underlying state the agent is currently in, the bottom scatter plots are colored based on the previous action the agent took. We initialized TSNE with the same random seed, with max iterations set to 1000, and perplexity set to 30. We present the median seeds for both cells **(top)** additive uses seed=55 and **(bottom)** multiplicative uses seed=67.

**Looking beyond performance:**

A natural question is why might the multiplicative cell perform significantly better than the other cells in this simple setting? One hypothesis is that the multiplicative cell does a better job at separating the histories on action sequence as compared to the additive operation. While this question is difficult to test, we can peer into the learned state of each cell and see if there are qualitative features that appear to help explain the better performance. To do this we take learned agents over different truncation values started using the same seed. After learning (using the same parameters as in Figure 4) we collect another 1000 steps of hidden states. With these hidden states we use TSNE (Van der Maaten and Hinton, 2008) to reduce the space of hidden states to two dimensions. The resulting scatter plots for the additive and multiplicative simple RNNs can be seen in Figure 7.

Overall, we observe the additive and multiplicative separate on the previous action equally well, matching our initial hypothesis. While action is important, the additive seems to be hyper-focused on action even as the cell is able to partition on environment state. The multiplicative, on the other hand, is able to cluster the hidden states for various environment states together with only minor separation on action as seen in states 1 and 7. It is possible this is a natural part of th learning process for both the cells, but the multiplicative is able to cluster the states in less samples. If we look at the median performer (seed=55 and seed=67 for the additive and multiplicative respectively) the additive fails to separate on environment state, while the multiplicative looks similarly to the previous seed.

## 5.2 Understanding when Action Encoding Does and Does Not Matter

In this section, we investigate learning behavior in two environments with slightly differing properties. The first domains is called TMaze (Bakker, 2002), depicted in Figure 3, with a size of 10, which was initially proposed to test the capabilities of LSTMs in RL using Q-Learning. The environment is a long hallway with a T-junction at the end. The agent receives an observation indicating whether the goal state is in the north position or south position at the T-junction (which is randomly chosen at the start of the episode). The agent can take actions in the compass directions. On each step the agent receives a reward of -0.1 and in the final transition receives a reward of 4 or -1 depending if the agent was able to remember which direction the goal was in. The agent deterministically starts at the beginning of the hallway. The observation in the first state is $[1, 1, 0]$ if the goal state is located above the agent and $[0, 1, 1]$ if the goal state is below the agent. In the final state of the hallway the agent receives $[0, 1, 0]$ as an observation, and everywhere else the observation is $[1, 0, 1]$.

Our control agents are constructed similarly to those used in the Ring World environment. The agent's network is a single recurrent layer followed by a linear layer. We perform a sweep over the size of the hidden state and learning rates, and selected all variants of a cell type to have the same value. We train our network over 300000 steps with further details reported in appendix F.2. We report the learned policy's performance over the final 10% of episodes by averaging the agent success in reaching the correct goal. We report our results using a box and whisker plot with the distribution. The upper and lower edges of the box represent the upper and lower quartiles respectively, with the median denoted by a line. The whiskers denote the maximum and minimum values, excluding outliers which are marked.

Shown in Figure 9 (left), all the cells have similar median performance with the GRU (with no action input) performing the best with the least amount of spread. This conclusion is the same across the size of the hidden state, where the multiplicative and factored variants performed poorly (see Appendix F for factored results). While this initially suggests the action embedding is not important beyond our simple Ring World experiment, notice the difference in how the environment's dynamics interact with the agent's action. In the TMaze, the underlying position of the agent is affected by only two of the actions (the East and West action), while the North and South actions only transition to a different state at the very end of the maze. Also, the agent's actions do not affect needs to be remembered, no matter what trajectory the agent sees the meaning of the first observation is always the same. Thus, these results are much less surprising. For example, the multiplicative variants will have to learn the update dynamics multiple times for the North and South actions.

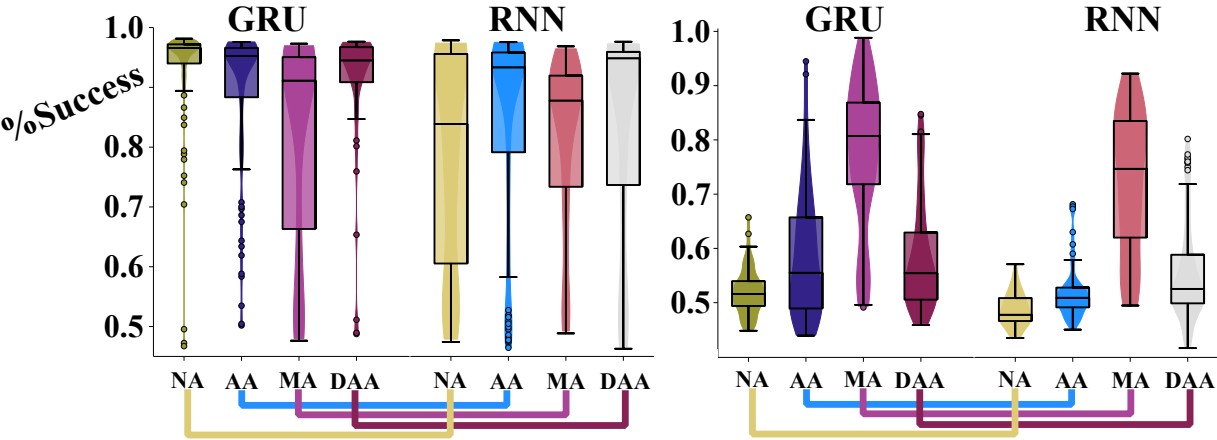

Figure 9: **(left)** Bakker's TMaze box plots and violin plots over the performance averaged over the final 10% with 50 independent runs. Trained over 300k steps with $\tau = 10$. All GRUs use a state size 6, while RNNs use a state size 20. The deep additive used an action encoding of $|\mathbf{a}| = 4$. **(right)** Directional TMaze comparison over the performance averaged over the final 10% of episodes with 100 independent runs trained over 300k steps with $\tau = 12$ for CELL (hidden size): RNN (30), AARNN (30), MARNN (18), DARNN (25, $|\mathbf{a}| = 15$), GRU (17), AAGRU (17), MAGRU (10), DAGRU (15, $|\mathbf{a}| = 8$).

To better replicate these dynamics in TMaze we add a direction component to the underlying state. For example, many robotics systems must be able to orient and turn to progress in a maze, which we hypothesize actions will be critical for modeling the state. The agent can take an action moving forward, turning clockwise, or turning counter-clockwise. Instead of the observations only being a function of the position, the agents direction plays a critical role. In the first state, the agent receives the goal observation $[1, 1, 0]$ when facing the wall corresponding to the goal's direction. All other walls have the observation $[0, 1, 0]$, and when not facing a wall the agent receives the observation $[0, 0, 1]$. In DirectionalTMaze the agent is forced to contextualize its observation by the action it takes before or after seeing the observation. We evaluate the state updates using the same settings as in the TMaze with results reported in Figure 9 (right).

Now that the agent must be mindful of its orientation, the action again becomes a critical component in learning. We see the multiplicative variants outperforming all other variants in this domain. Without action, the GRU and RNN are unable to learn, and even the additive and deep additive versions are unable to learn in 300000 steps. We also sweep over the number of factors and report the performance compared to the multiplicative and additive variants as shown in Figure 10. We found that as the factors increase, generally the performance

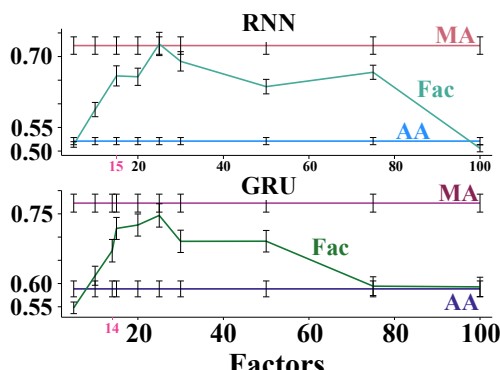

Figure 10: Sensitivity curves over number of factors $M$ with standard error for the **(top)** FacRNN (30) and **(bottom)** Fac-GRU (17). All agents were trained over 300k steps. See Appendix F.3 for sweeps over different state sizes. We use the data generated by a sweep over the learning rate with 40 runs and compare to the data in figure 9. The red labels on the x-axis indicate when the network has the same number of parameters as the multiplicative.

increases as well. This matches our expectations, as with increased factors the factored variants should better approximate the multiplicative variances. But there is a tradeoff when adding too many factors, causing performance to decrease substantially. While the factored variant has some interesting properties, we decide to focus the remaining experiments using the base architectures (NA, MA, AA, DA) and report full results with the factored variant in Appendix F.

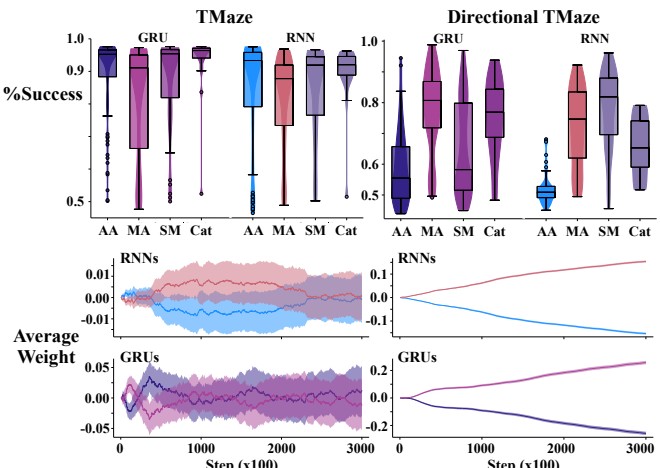

Figure 11: Two variants of combining cells. State size chosen based on procedures of previous environments. (**top**) Performance of success rates (**left**) TMaze with same basic parameters as above for CELL (hidden size): Softmax GRU (6), Cat GRU (6), Softmax RNN (20), Cat RNN (20). (**right**) Directional TMaze with same parameters as above for CELL (hidden size): Softmax GRU (8), Cat GRU (12), Softmax RNN (15), Cat RNN (22). (**Bottom**) Average softmax weights of cells over training with standard error over runs.

### 5.2.1 Combining Cell Architectures

In this section, we consider the effects of combining the additive and multiplicative cells through two types of combination techniques. We see these architectures as a minor step toward building an architecture which learns the structural bias currently hand designed.

We combine the hidden state between an additive and multiplicative operation through two techniques. The first is through an element-wise softmax. Both the additive and multiplicative have the same size hidden state ($\mathbf{s}^a$ and $\mathbf{s}^m$ respectively), and each element of the hidden states are weighted by

$$\mathbf{s}_i = \frac{e^{\theta_i^a}\mathbf{s}_i^a + e^{\theta_i^m}\mathbf{s}_i^m}{e^{\theta_i^a} + e^{\theta_i^m}}$$

where $\boldsymbol{\theta}^a, \boldsymbol{\theta}^m \in \mathbb{R}^n$. This should learn which cell to use depending on the structure of the problem. The second combination is through concatenating the two hidden state together $\mathbf{s} = cat(\mathbf{s}^a, \mathbf{s}^m)$. This gives more room for experts to add more state to the different architectures, but in this work we fix the two architectures to have the same state size.

We compare these combinations to the original architectures in TMaze and Directional TMaze following the same procedure as above. We expect these cells to perform as well as either the additive or the multiplicative (which ever is doing the best in the specific domain). The results can be seen in Figure 11. Overall, the softmax combination performs similarly or slightly better than the multiplicative version except in the Directional TMaze for the GRUs. In TMaze, concatenating the two states together performed better than the additive and multiplicative cells, but this operation worked slightly worse than the multiplicative in the Directional TMaze. To test the hypothesis that the softmax weighting should emphasize the better cell in a given domain we show the softmax weighting over the training period. For the TMaze the weightings end being approximately equivalent while the Directional TMaze shows a very distinct separation where the multiplicative is weighted significantly more and the additive is continually down-weighted.

### 5.3 Learning State Representations from Pixels

Finally, we perform an empirical study in two environments with non-binary observations. We are particularly interested in whether the recurrent architectures perform comparably when the observation needs to be transformed by fully connected layers, or when the observation is an image. We only use the GRU cells in these experiments. Full details can be found in Appendix F.

The first domain we consider is a version of DirectionalTMaze which uses images instead of bit observations. The agent receives a gray scale image observation on every step of size $28 \times 28$. The agent sees a fully black screen when looking down the hallway, and a half white half black screen when looking at a wall. The agent observes an even (or odd) number sampled from the MNIST (LeCun et al., 2010) dataset when facing the direction of (or opposite of) the goal. The rewards are -1 on every step and 4 or -4 for entering the correct

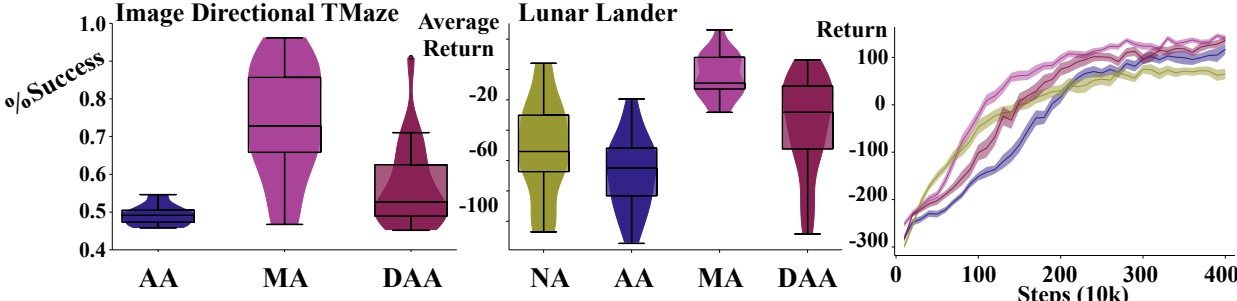

Figure 12: **(left)** Image Directional TMaze percent success over the final 10% of episodes for 20 runs for CELL (hidden size): AAGRU (70), MAGRU (32), DAGRU (45, |**a**| = 128). Using ADAM trained over 400k steps, $(\tau) = 20$. GRU omitted due to prior performance. **(center)** Lunar Lander average reward over all episodes for CELL (hidden size): GRU (154), AAGRU (152), MAGRU (64), DAGRU (152, |**a**| = 64) and $(\tau) = 16$. **(right)** Lunar Lander learning curves over total reward. Ribbons show standard error and a window averaging over 100k steps was used. Lunar Lander agents were trained for 20 independent runs for 4M steps.

and incorrect goal position respectively. We report the same statistic as in the prior TMaze environments, with the environment size set to 6. Notice the hallway size is smaller and the negative reward is larger, this was to speed up learning for all architectures.

Results for the Image DirectionalTMaze can be seen in Figure 12. In this domains, the multiplicative performs quite well, although not as well as in the simple version. The AAGRU is unable to learn in this setting, and the deep additive variant performs slightly better than the additive.

### 5.4 Learning State Representations from Agent-Centric Sensors

The second domain is a partially observable version of the LunarLander-v2 environment from OpenAI Gym Brockman et al. (2016). The goal is to land a lander on the moon within a landing area. Further details and results can be found in Appendix F.5. To make the observation partially we remove the anglular speed, and we filter the angle $\theta$ such that it is 1 if $-7.5 \leq \theta \leq 7.5$ and 0 otherwise. We report the average reward obtained over all episodes, and learning curves.

As seen in Figure 12, our findings generalize to this domain as well. The multiplicative variant improves over the factored (see Appendix F, additive, and deep additive variants significantly. In the LunarLander environment the multiplicative learns faster, reaching a policy which receives on average 100 total reward per episode. Both the additive and factored eventually learn similar policies, while the standard GRU seems to perform less well (although not statistically significant from the additive variant). The average return is 100 less than some of the best agents on this domains. When we look at the individual median curves we see the agent does this well 50% of the time (see Appendix F). This difference can be explained by the failure start states being more frequent than in the fully observed case.

## 6 Open Problems for Recurrent Architectures in RL

Recurrent architectures are often taken off-the-shelf from the supervised learning setting for use in reinforcement learning. While this has been moderately successful, the RL problem poses challenges not often considered by supervised learning. Below we discuss three interesting properties of an RL system, and how they affect learning using recurrent networks.

**Practical Online Recurrent Learning:**

In reinforcement learning, it is desirable to learn as much about the most recent experience before selecting an action (i.e. to learn online and incrementally). Learning efficiently online enables adapting behavior in real time and scaling to massive data-streams and architectures. This puts pressure on the learning

system to update the weights within a set amount of time so the system can act (Sutton et al., 2011; White, 2015), which is often not a concern in the supervised setting. In settings where an agent must move around its environment independently, the on-board computational system can be heavily constrained by the power of the processor as well as limited energy from the battery. An algorithm whose computational and memory complexity scales independently of the sequence length (without the quadratic complexity on size of the network as real time recurrent learning (Williams and Zipser, 1989)) and could be applied online-incrementally would be a major breakthrough in using recurrent architectures for RL and computationally constrained systems generally. A detailed discussion on relevant literature is in Appendix A.

**Active Data Collection Matters:**

Imagine an agent in a hallway with recognizable observations only at the beginning and end of the hallway, much like our TMaze environments. The agent must learn a state update which spans at least the length of the hallway. But this is in the best case scenario when the agent prioritizes making it to the end of the hallway. In reality, our agents will randomly explore the hallway until the end, often extending the length of the sequence the agent needs to learn over. The interaction between the agent's behavior (or exploration) and the difficulty of training under partial observability with a recurrent agent is currently unexplored. Active data collection strategies could mitigate the length of long-temporal dependencies, which would show massive improvements in our agent's learning efficiency and ability.

To show the potential of active data collection, we will briefly revisit the Directional TMaze. Start with an agent who has learned the base task of the Directional TMaze environment. If we force the agent to take specific actions and to start in a specific orientation at the beginning of the episode, we could feasibly teach the agent to artificially extend the horizon of its policy without increasing the length of the training sequence. See Figure 13 for preliminary results of how behavior can extend the horizon of the agent's policy. In this experiment, trained multiplicative agents are paced through a set of interventions. The naive strategy uses epsilon greed after forcing the agent to step forward twice down the hallway. The hand designed sequence, instead guides the agent through a series of forced actions to build to the final desired policy. This simple experiment shows the potential for slowly extending the temporal horizon of a policy without adjusting the truncation value by intervening on the agent's behavior.

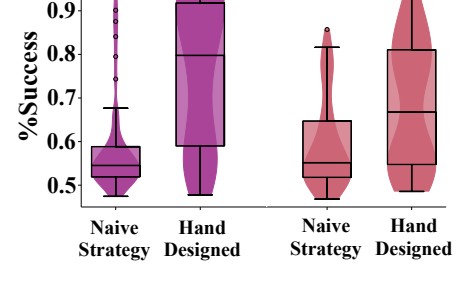

Figure 13: Average success over the intervention taking the go forward activation and starting in the eastward position. **(Naive Strategy)** Using the evaluated intervention over 60k steps for training, **(Hand Designed)** a sequence of hand designed interventions to build up to the final evaluation intervention over 60k steps.

**Insight Beyond Learning Curves:**

Learning curves provide little understanding of an agent's learning process and this likely limits algorithmic progress in partially observable settings. Unfortunately, such metrics can't be used to address more complicated questions about the agent and its behavior. While searching for SOTA is admirable, deeper questions about the internal learned structures and behaviors of our agents are often, but not always, ignored. Analyzing the internal dynamics of an agent with recurrent architectures is uniquely challenging in reinforcement learning. Some challenges include (see Appendix B for details):

- Generating data for evaluating and analyzing representation learning is an especially difficult problem for agents with recurrence. The data generated must be coherent trajectories the agent may potentially experience in the environment, meaning data generating policies must be selected to provide coverage over the space of agent-environment interactions.

- Current tools for analyzing state representations are designed for NLP (Karpathy et al., 2015; Ming et al., 2017) and are ill-suited for analyzing the link between the environment, the agent's state, and the behavior policy.

- Analyzing the behavior of our agents through performance metrics leaves many questions unanswered: How is the agent behaving to solve the task? When does the agent make a long-term decision? In what circumstances might the agent's policy fail?

## 7   Conclusion

In this paper, we empirically evaluated several strategies for incorporating the previous action into the state update of a recurrent neural network. We demonstrated in several environments from several observation types that this choice can have a large impact on an RL agent's performance for both prediction and control. Our empirical results suggest that the multiplicative operation performs the best even when using a smaller state vector, and the factored and the deep additive versions perform marginally better than the additive versions in most domains.

While the multiplicative seems to be the clear winner on the tested domains, it is important to note not all domains require this architecture. One interesting strategy could be to use the softmax combined cells to decide which cell to use in your final architecture (by looking at the softmax weighting). One could also imagine an architecture which is able to learn which cell to use conditioned on the history of the agent (see Section E.1). Until better architectures for RL are defined this choice is left to system designers. While the additive and deep additive versions under-performed compared to the other encodings, it still out-performed naively using RNNs without action input.

What is apparent in our experiments here and empirical evidence recently gathered on the performance of recurrent architectures in the online setting (Rafiee et al., 2022; Schlegel et al., 2021) is that the methods and architectures developed and utilized by supervised learning might not be suitable for the reinforcement learning problem. This paper uncovered a simple choice can have a large impact, and provides some evidence that the assumptions made in supervised learning might be holding back recurrent architectures in the reinforcement learning setting (see Appendix C for details). Small, focused studies using recurrent agents in controlled experiments will continue to produce insights on the limitations of the base algorithms and continue to inspire future algorithm developments.

### Acknowledgments

We would like to thank the Alberta Machine Intelligence Institute, IVADO, NSERC and the Canada CIFAR AI Chairs Program for the funding for this research, as well as Compute Canada for the computing resources used for this work. We would also like to acknowledge the anonymous reviewers whose comments have made the paper more clear, and Khurram Javed for his suggestions which lead to the Image Directional TMaze environment.

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

# A    Learning Long-Temporal Dependencies from Online Data

Learning long-temporal dependencies is the primary concern of both RL and SL applications of recurrent networks. While great work has been done to coalesce around a few potential architectures and algorithms for SL settings, these are often found lacking in the online-incremental RL context (Sodhani et al., 2019; Rafiee et al., 2022; Schlegel et al., 2021) discussed in section 6. Not only do agents need to learn from the currently stored data (i.e. in an experience replay buffer), they must also continually incorporate the newest information into their decisions (i.e. update online and incrementally). The importance of learning state from an online stream of data has been heavily emphasized in the past through predictive representations of state (Littman and Sutton, 2002), temporal-difference networks (Sutton and Tanner, 2005) and GVF networks (Schlegel et al., 2021), and in modeling trace patterning systems (Rafiee et al., 2022). From a supervised learning perspective, several problems like saturating capacity and catastrophic forgetting are cited as the most pressing for any parametric continual learning system (Sodhani et al., 2019). Below we suggest a few alternative directions needing further exploration in the RL context.

The current standard in training recurrent architectures in RL is truncated BPTT. This algorithm trades off the ability to learn long-temporal dependencies with computation and memory complexity. Currently, the system designer must set the length of temporal sequences the agent needs to model (as would be needed for truncated BPTT to be effective (Mozer, 1995; Ke et al., 2018; Tallec and Ollivier, 2018; Rafiee et al., 2022)). Setting this length is a difficult task, as it interacts with the underlying environment and the agent's exploration strategy (see section 6 for more details). As the truncation parameter increases it is known that the gradient estimates become wildly variant (Pascanu et al., 2013; Sodhani et al., 2019), which can make learning slow.

An alternative to (truncated) BPTT is real time recurrent learning (RTRL) (Williams and Zipser, 1989). Unfortunately RTRL is known to suffer high computational costs for large networks. Several approximations have been developed to alleviate these costs (Tallec and Ollivier, 2018; Mujika et al., 2018), but these algorithms often struggle from high variance updates making learning slow. The approximation to the RTRL influence matrix proposed by Menick et al. (2020) shows significant promise in sparse recurrent networks, even outperforming BPTT when trained fully online. Ke et al. (2018) propose a sparse attentive backtracking credit assignment algorithm inspired by hippocampal replay, showing evidence the algorithm has beneficial properties of both BPTT and truncated BPTT. The focused architecture was often able to compete with the fully connected architecture on length of learned temporal sequence and prediction error on several benchmark tasks. Another line of search/credit assignment algorithms is generate and test (Kudenko and Hirsh, 1998; Mahmood and Sutton, 2013; Dohare et al., 2021; Samani and Sutton, 2021). These search algorithms aren't as tied to their initialization as other systems as they intermittently inject randomness into their search to jump out of local minima. Many of these approaches combine both gradient descent and generate and test to gain the benefits of both. While a full generate and test solution is possible, finding the right heuristics to generate useful state objects quickly could be problem dependent.

Learning long-temporal dependencies through regularizing objectives on the state has shown promise in alleviating the need for unrolling the network over long-temporal sequences. Schlegel et al. (2021) use GVFs to make the hidden state of a simple RNN predictions about the observations showing potential in lightening the need for BPTT. This approach is sensitive the GVF parameters to use as targets on the state of the network. Predictive state recurrent neural networks (Downey et al., 2017) combine the benefits of RNNs and predictive representations of state (Littman and Sutton, 2002) in a single architecture. They show improvement in several settings, but don't explore the model when starved for temporal information in the update. Another approach is through stimulating traces, as shown by Rafiee et al. (2022), where traces of observations are used to bridge the gap between different stimuli. Instead of traces, an objective which learns the expected trace (van Hasselt et al., 2021) of the trajectory could provide similar benefits as a predictive objective. One can even change the requirements on the architecture in terms of final objectives. Mozer (1991) propose to predict only the contour or general trends of a temporal sequence, reducing the resolution considerably. Value functions are another object which takes an infinite sequence and reduces resolution to make the target easier to predict (Sutton, 1995; Sutton et al., 2011; Modayil et al., 2014; van Hasselt and Sutton, 2015).

It is also possible to reduce or avoid the need for BPTT for modeling long-temporal sequences by adjusting the internal mechanisms of the recurrent architecture. Echo-state Networks (Jaeger, 2002) are one possible direction. Related to the generate and test idea, echo-state networks rely on a random fixed "reservoir" network, where predictions are made by only adjusting the outgoing weights. Because the recurrent architecture is fixed, no gradients flow through the recurrent connections meaning no BPTT is needed to estimate the gradients. Unfortunately, these networks are dependent on their initializations making them hard to deploy in practice. Mozer (1995) propose a focused architecture design, where recurrent connections are made more sparsely (even just singular connections). This significantly reduces the computational complexity of RTRL and allows for a focused version of BPTT.

Transformers (Vaswani et al., 2017) are a widely used alternative to recurrent architectures in natural language processing. Transformers have also shown some success in reinforcement learning but either require the full sequence of observations at inference and learning time (Mishra et al., 2018; Parisotto et al., 2020) or turn the RL problem into a supervised problem using the full return as the training signal (Chen et al., 2021). Because of these compromises, it is still unclear if transformers are a viable solution to the state construction problem in continual reinforcement learning.

## B   Insight Beyond Learning Curves

Learning curves showing the agent's performance, usually through episodic return or prediction error, over the agent's lifetime has been the primary method algorithms are compared. Unfortunately, such metrics can't be used to address more complicated questions about the agent and its behavior. While searching for SOTA is admirable, deeper questions about the internal learned structures and behaviors of our agents are often, but not always, ignored. Analyzing the internal dynamics of an agent with recurrent architectures is uniquely challenging in reinforcement learning.

One challenge is how data is generated. Unlike SL whose data is usually a dataset designed ahead of time, RL generates data through interactions with an environment whose underlying dynamics are likely unaccessible to the system designer. While randomly generated data, in combination with tools from NLP (Karpathy et al., 2015; Ming et al., 2017), can give us some insight into how our agent's perform, see section 5.1, extending the analysis to larger domains could leave large parts of the agent-environment interactions unseen.

While we provide some representational analysis in the prediction setting, further results in the control setting would be even more beneficial. Unfortunately, many of the analysis tools we considered require the "correct" target at a given time. In the control setting, even when the underlying dynamics of an environment can be fully specified (say in lunar lander) a notion of what the right action is at a given time can be extremely difficult to discover. Future work should go into analyzing the link between histories, agent state, environment state, and behavior.

Even when analyzing the behavior of our agent, using the performance metric as the primary measure is deeply flawed. This type of analysis fails to address questions such as: How the agent might be behaving? When does the agent make a long-term decision? In what circumstances might the agent's policy fail? Analyzing the agent as a non-linear coupled system with the environment through a series of dynamical equations could lead to further insight on conditions which lead the agent to behave in certain ways or when certain decisions are made by the agent. Beer (2003) develop a series of questions and experiments to analyze an artificial agent from this perspective in a simple catcher like domain. While these tools would be difficult to apply to real-world problems, using them in simulations could provide useful insight into a full description of the agent's learned policy.

Because of the above challenges there are several lingering questions about these types of agents left unanswered in these domains. 1) Under what conditions will the agent's policy fail in an environment? 2) How robust are the policies to out-of-distribution events and how does this effect the hidden state? 3) What algorithms do the learning process discover to solve the domains reliably? 4) Is the model stable over a long training period or in a continual domain? 5) When does the agent make a decision, and does the agent

stick to this decision? We believe answering these questions and more can lead to better understanding of recurrent agents as well as pathways to better algorithms for training such agents.

## C  Architectural Choices

Below are several architectural choices we made which should be empirically explored in future work:

**Cell architecture:** As exemplified in this paper, the architectural choices made in the supervised learning setting may not be the best suited for the RL setting. Here we focused on one simple architectural choice, how to incorporate action into the state, but show sometimes massive improvements in the networks ability to predict and control. Sections 5.2.1 and E.1 explore this further, but future work should investigate novel RL based cell types.

**The woes of the experience replay buffer:** Current deep learning, including recurrent architectures, in reinforcement learning include the need for an experience replay buffer. While a learning algorithm which overcomes this limitation would likely be preferable, in the short term cohesive strategies for combining an experience replay with recurrent architectures should be empirically explored. There are two major approaches currently: 1) using the stale traces, or 2) warming up the agent from the beginning (or some number of time steps prior) of an episode (Hausknecht and Stone, 2015). We use a third strategy here (using gradient information to refresh the hidden state to minimize the objective), but found little difference between this and the stale approach. For much more insight and discussion on this choice see Kapturowski et al. (2018).

**Target networks and state:** How should we initialize the hidden state for a target network? In this paper, we used the state stored (in the experience replay buffer) for the main model.

**To continually learning, or to not:** The trade-offs between a continually learning and non-continually learning agent is extremely important for both recurrent and feed-forward architectures. In this paper, we chose to report results when the agent is still learning, but only on stationary domains. Any learning system should track the environment as it changes, or as the agent experiences novel states. Recurrent architectures have an added problem that the hidden state is an accumulation of the agent's entire history, meaning novel observations could irreparably harm the hidden state if it is not reset at regular intervals (i.e. in an episodic domain). This will potentially harm the agent's performance for its entire lifetime if it is unable to adjust it's weights accordingly. This is not the case in a feed-forward network, where novel observations will not effect the long-term behavior of an agent in known parts of the state space. This effect is understudied in the literature for recurrent agents, but is an important aspect of deploying recurrent RL agents in real world systems.

**Objectives matter:**  It is known that some objective functions are more learnable in both the fully and partially observable settings (Mozer, 1991; van Hasselt and Sutton, 2015). Auxiliary tasks are often also used to augment the networks objective function (Jaderberg et al., 2017), or to constrain the learned state (Schlegel et al., 2021). Which objective functions, auxiliary tasks, and learning algorithms are more best when applied to training a recurrent networks?

## D  Some background on tensors

When introducing the multiplicative update and the factored update, we relied on the weight matrix being a 3-tensor rather than a matrix of weights. While this make the notation convenient in the main paper, it requires some atypical background information. This section provides some background on tensors and some low-rank decompositions of tensors.

The simplest, albeit slightly inaccurate, way to describe and use a tensor is as a multi-dimensional array of numbers (either real or complex) which transform under coordinate changes in predictable ways. In this paper, we consider tensors as multi-dimensional arrays using Einstein summation notation. The ith, jth, kth component of an order-3 tensor will be denoted with lower indices $\mathbf{W}_{ijk} \in \mathbb{R}$ with associated dimension size denoted with corresponding uppercase letters as $\mathbf{W} \in \mathbb{R}^{I \times J \times K}$.

Like matrices, tensors have a number of decompositions which can prove useful. For example, every tensor can be factorized using canonical polyadic decomposition (CP decomposition), which decomposes an order-N tensor $\mathbf{W} \in \mathbb{R}^{I_1 \times I_2 \times \ldots \times I_N}$ into N matrices as follows

$$
\begin{aligned}
\mathbf{W}_{i_1,i_2,\ldots} &= \sum_{r=1}^{R} \lambda_r \mathbf{W}_{i_1,r}^{(1)} \mathbf{W}_{i_2,r}^{(2)} \ldots \mathbf{W}_{i_N,r}^{(N)} \\
&= \lambda_r \mathbf{W}_{i_1,r}^{(1)} \mathbf{W}_{i_2,r}^{(2)} \ldots \mathbf{W}_{i_N,r}^{(N)} \quad \triangleright \text{Explicit summation over } r \in \{1,\ldots,R\}.
\end{aligned}
$$

where $\mathbf{W}^{(j)} \in \mathbb{R}^{I_j \times R}$, and $R$ is the rank of the tensor. This is a generalization of matrix rank decomposition, and exists for all tensors with finite dimensions.

Working with tensors takes a bit more care in deciding which fibers (generalization of row and column) the product should be over. One type of product is known as the n-mode product which is defined as follows

$$
(\mathbf{W} \times_n \mathbf{v})_{i_1,i_2,\ldots,i_{n-1},j,i_{n+1},\ldots i_N} = \mathbf{W}_{i_1,i_2,\ldots,i_{n-1},i_n,i_{n+1},\ldots i_N} \mathbf{v}_{j,i_n}
$$

where $\mathbf{v} \in \mathbb{R}^{J,I_n}$. An important property which will be used in this paper is some simplifications we can make when considering n-mode products with their rank decomposition. For simplicity we will simplify an order-3 tensor ($\mathbf{W} \in \mathbb{R}^{IJK}$, $\mathbf{W}_{ijk} = \lambda_r a_{ir} b_{jr} c_{kr}$, $\mathbf{v}^M = \mathbf{v}^{(1,M)} \in \mathbb{R}^{1 \times M}$),

$$
\begin{aligned}
(\mathbf{W} \times_2 \mathbf{v}^J \times_3 \mathbf{v}^K)_{i,1,1} &= \sum_{k=1}^{K} \left( \sum_{j=1}^{J} \mathbf{W}_{ijk} \mathbf{v}_{1j}^J \right) \mathbf{v}_{1k}^K \\
&= \sum_{k=1}^{K} \sum_{j=1}^{J} \left( \sum_{r=1}^{R} \lambda_r a_{ir} b_{jr} c_{kr} \right) \mathbf{v}_{1j}^J \mathbf{v}_{1k}^K \\
&= \sum_{r=1}^{R} \lambda_r a_{ir} \left( \sum_{j=1}^{J} b_{jr} \mathbf{v}_{1j}^J \right) \left( \sum_{k=1}^{K} c_{kr} \mathbf{v}_{1k}^K \right) \\
&= \sum_{r=1}^{R} \lambda_r a_{ir} \left( \mathbf{v}^J \mathbf{B} \odot \mathbf{v}^K \mathbf{C} \right)_{1r} \\
\mathbf{W} \times_2 \mathbf{v}^J \times_3 \mathbf{v}^K &= \boldsymbol{\lambda} \mathbf{A} \left( \mathbf{v}^J \mathbf{B} \odot \mathbf{v}^K \mathbf{C} \right)^{\top} \quad \triangleright \boldsymbol{\lambda}_{i,i} = \lambda_i
\end{aligned}
$$

Similarly to CP decomposition, Tucker rank decomposition can be used to create a similar operation. Tucker rank decomposition decomposes an order-N tensor $\mathbf{W} \in \mathbb{R}^{I_1 \times I_2 \times \ldots \times I_N}$ into N matrices another order-N tensor $G \in \mathbb{R}^{R_1 \times R_2 \times \ldots \times R_N}$ as follows

$$
\mathbf{W}_{i_1,i_2,\ldots i_N} = \sum_{r_1=1}^{R_1} \sum_{r_1=1}^{R_1} \cdots \sum_{r_1=1}^{R_1} g_{r_1 r_2 \ldots r_N} \mathbf{W}_{i_1,r_1}^{(1)} \mathbf{W}_{i_2,r_2}^{(2)} \cdots \mathbf{W}_{i_N,r_N}^{(N)}.
$$

With similar simplifications to CP decomposition,

$$
\begin{aligned}
(\mathbf{W} \times_2 \mathbf{v}^J \times_3 \mathbf{v}^K)_{i,1,1} &= \sum_{k=1}^{K} \left( \sum_{j=1}^{J} \mathbf{W}_{ijk} \mathbf{v}_{1j}^J \right) \mathbf{v}_{1k}^K \\
&= \sum_{k=1}^{K} \sum_{j=1}^{J} \left( \sum_{p=1}^{P} \sum_{q=1}^{Q} \sum_{r=1}^{R} g_{pqr} a_{ip} b_{jq} c_{kr} \right) \mathbf{v}_{1j}^J \mathbf{v}_{1k}^K \\
&= \sum_{p=1}^{P} \sum_{q=1}^{Q} \sum_{r=1}^{R} g_{pqr} a_{ip} \left( \sum_{j=1}^{J} b_{jq} \mathbf{v}_{1j}^J \right) \left( \sum_{k=1}^{K} c_{kr} \mathbf{v}_{1k}^K \right) \\
&= \sum_{p=1}^{P} \sum_{q=1}^{Q} \sum_{r=1}^{R} g_{pqr} a_{ip} \left( \mathbf{v}^J \mathbf{B} \right)_{1q} \left( \mathbf{v}^K \mathbf{C} \right)_{1r} \\
\mathbf{W} \times_2 \mathbf{v}^J \times_3 \mathbf{v}^K &= G \times_1 \mathbf{A}^\top \times_2 \left( \mathbf{v}^J \mathbf{B} \right)^\top \times_3 \left( \mathbf{v}^K \mathbf{C} \right)^\top \\
&= \mathbf{A} \left[ \left( G^\top \times_2 \left( \mathbf{v}^J \mathbf{B} \right)^\top \right) \left( \mathbf{v}^K \mathbf{C} \right)^\top \right].
\end{aligned}
$$

One interesting property of this operation is now each of the dimensions can have a separately tuned rank, giving the system designer more discretion on where to focus representational resources.

Using a lower rank approximation of a multiplicative operation has been derived before several times. A multiplicative update was used to make action-conditional video predictions in Atari (Oh et al., 2015). This operation also appears in a lower-rank approximation defined by Predictive State RNN hidden state update (Downey et al., 2017), albeit never performed as well as the full rank version. We find similarly that both factorizations perform below the full tensor version (i.e. the multiplicative). We don't report results for the Tucker rank decomposition as it performed similarly to the CP decomposition.

# E  Further Investigations

## E.1  Learning the structural bias

So far, we've focused on architectures which have static architectures, where the agent has no agency in learning the appropriate structure. While this strategy seems to be reasonable as a starting point, in the future an architecture which can learn these different networks would be more desirable. We propose one such architecture here and an initial empirical evaluation of this architecture, leading to a discussion on the problematic properties such an algorithm might have.

$$
\begin{aligned}
z_t^i &= f_{\text{update}}(\boldsymbol{\theta}, \mathbf{x}_t, \mathbf{a}_{t-1}) \\
\psi_t &= f_{\text{GN}}(\mathbf{x}_t, \mathbf{a}_{t-1}) \\
\mathbf{s}_t &= z_t \odot psi_t.
\end{aligned}
$$

Where $f_{\text{GN}} : \mathcal{A} \times \mathcal{S} \times \mathcal{O} \to \mathbb{R}^{|\mathbf{s}|}$ is a parameterized function which is used to create a mixture over the experts state $z \in \mathbb{R}^{|\mathbf{s}|}$ produced by a state update function $f_{\text{update}}$. Both the gating function and the expert rnn state update function can be arbitrarily constructed. In this section we focus on the simple RNN update and a feed forward ANN with relu activations and a softmax activation on the final layer.

The results are presented in Figure 14. A sweep over various number of experts and a simple gating network with a single layer and softmax activation. As compared to the additive and multiplicative the mixture of experts RNN network performs in-between the two networks. The GRU, on the other hand, fails to perform well in this domain. This might be related to the results seen in section 5.2.1, where both the combined GRUs failed to outperform the multiplicative.

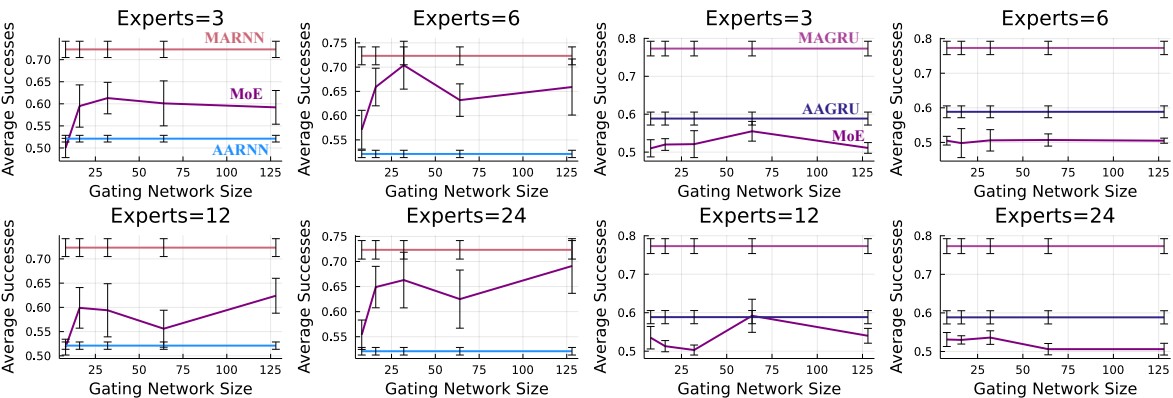

Figure 14: Directional TMaze sweep over size of the gating network (i.e how many units in a single hidden layer with relu activations and an output network w/ softmax output) for (**left**) RNN and (**right**) GRU. All experiments follow the procedures from previous results, except the MoE networks use 20 runs.

## E.2 TSNEs over Time

In section 5.1, we hypothesized the separation of action faced by the additive agent could have been an artifact of the learning dynamics. To test this hypothesis we created TSNEs for various number of steps in the environment for both the additive and multiplicative. The results can be seen in figure 15. For the multiplicative we choose [50000, 75000, 100000, 300000] which shows the major learning milestones of the network. For the additive we choose [50000, 150000, 200000, 500000] which goes beyond the original experiment's time limit and shows the major milestones when the network separates the histories according to state. For 100000 steps of training for the multiplicative we can see similar properties where the actions taken to get to specific states are quite separated. As the number of samples grow, to 300000, we see the states converging to be mostly clustered together regardless of the action taken. The additive version never sees the states converging, where even after 500000 timesteps the actions are still regarded highly by the network.

## E.3 Online Setting

In this section, we test to see if our conclusions from the previous sections generalize to the fully online setting. We report some results for Ring World and DirectionalTMaze here, with further results in appendix F.1 and F.3 respectively. For both environments, all applicable settings are the same as in the replay counter parts. The only difference is in how the network is updated. Instead of sampling from an experience replay, we store a history of the truncation length and update the network on every step using the same semi-gradient updates.

We present the results for the online setting in figure 16. Compared to the replay setting, we can see all the variants performed worse across the board. For DirectionalTMaze the AAGRU and MAGRU have a reasonable median performance. The MARNN and FacGRU are the only other cells which have runs reaching good performance, but overall perform poorly. We expect initialization plays a large role in the networks performance and should be investigated. We also see similar trends in Ring World, except the RNN variants outperform the GRUs. Another interesting consequence in the online setting, is the need to increase the truncation value and hidden state size to perform reasonably for both DirectionalTMaze and Ring World.

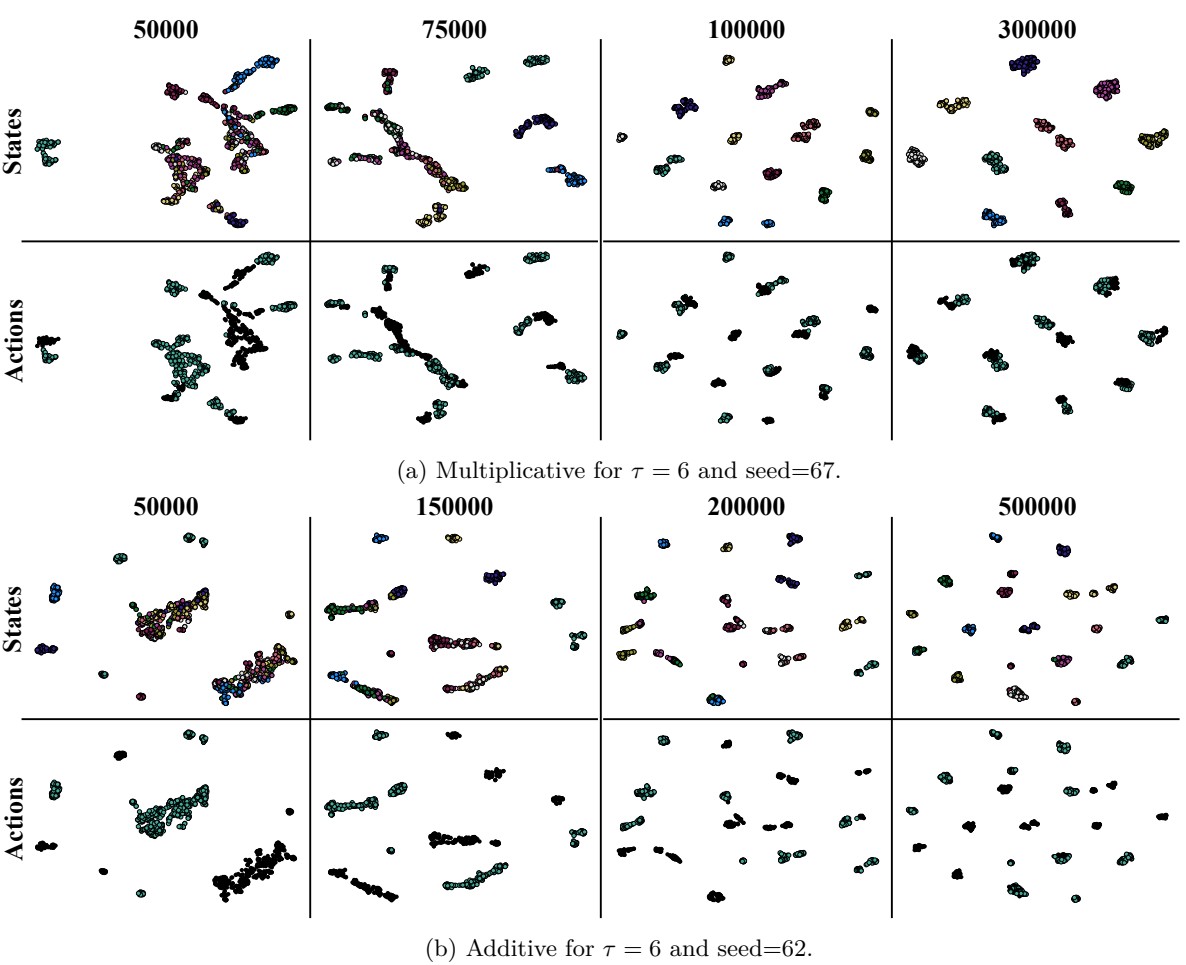

(a) Multiplicative for $\tau = 6$ and seed=67.

(b) Additive for $\tau = 6$ and seed=62.

Figure 15: TSNE plots for multiplicative and additive RNNs for various number of training samples.

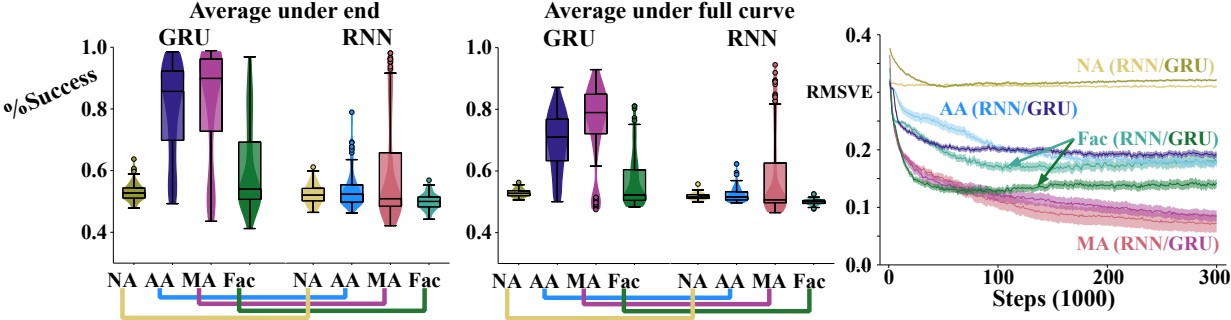

Figure 16: Online: **(left + middle)** Directional TMaze percent success in reaching the goal over the final 10% of episodes with 100 independent runs for CELL (hidden size): RNN (46), AARNN (46), MARNN (27), FacRNN (46) $M = 24$, GRU (26), AAGRU (26), MAGRU (15), FacGRU (26) $M = 21$. **(right)** Ring World learning curves over RMSVE with 100 independent runs for: RNN (20), AARNN (20), MARNN (15), GRU (12), AAGRU (12), MAGRU (9). Ribbons show standard error and a window averaging over 10k steps was used. Factored variants were excluded for clarity, due to high variance results. All agents were trained over 300k steps.

## E.4 Masked Grid World

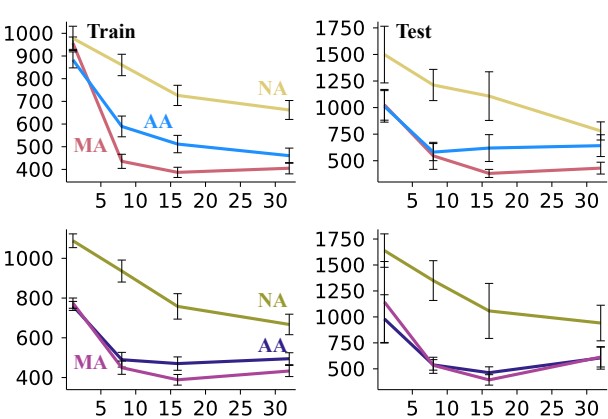

Figure 17: Average number of steps to goal over truncation for Masked Grid World **left** over the entire learning process **right** from a set of representative states after training. Cell (state size) **top** RNN (24), AARNN (20), MARNN (10) **bottom** GRU (24), AAGRU (20), MAGRU (10).

While the TMaze and DirTMaze give some insight into when different encodings might be preferable, the DirTMaze and Ring World share similar dynamics in how the actions effect the unobserved state of the MDP. Specifically, there are two actions which effect a state component symmetrically. This prompts the question on whether this property is driving the benefits of the multiplicative update's success, or whether there are other scenarios where the multiplicative does better. We propose a new environment which is a simple grid world with border wrapping. The agent can take a step in all the cardinal directions, and observes when it enters a random subset of the states (all aliased together). The goal state is also randomly selected at the beginning of an agent's life. This creates random action observation patterns the agent must notice and act on to get to the goal. The border wrapping prevents the agent from moving to a corner of the environment and then going to the goal.

In figure 17, we confirm the hypothesis that the improvement with multiplicative update can be meaningful even when the state-action sequences are randomly placed in the environment. While the improvement is much less drastic than the Ring World and DirTMaze, the improvement is still significant with standard error bars. Another interesting observation is the difference matters much more for the simple recurrent update than the GRU.

## E.5 Further Results for the Deep Additive and Deep Multiplicative Architectures

We provide two more experiments with results reported in figure 18. First we provide results over various action encoding sizes for the Directional TMaze environment using the Deep Additive network from the main paper. Overall, we found the size of the encoding network to not make a large difference in the final performance. In effect, this result suggests there is still a core limitation with the deep additive operation

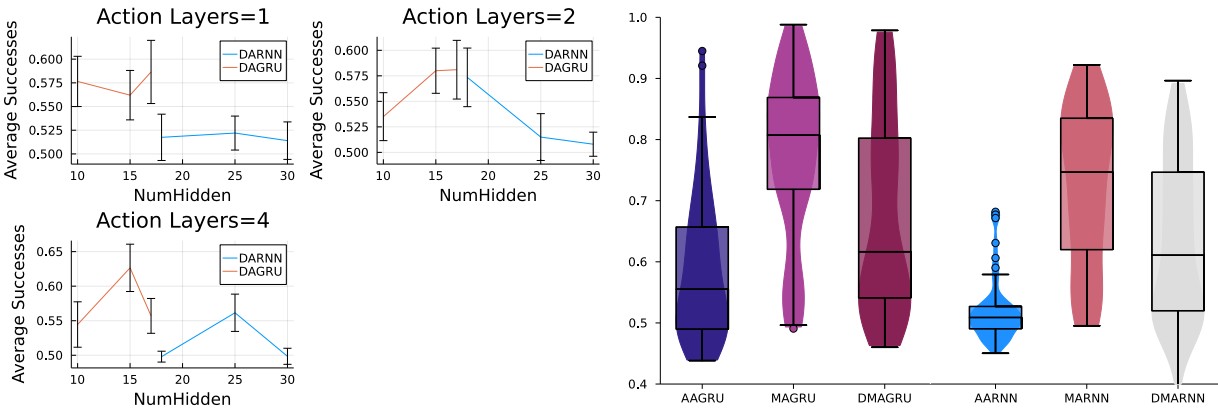

Figure 18: **(left)** Resulting average success over the last 10% of episodes for various number of hidden units in the action encoding network for the deep additive networks with standard error intervals. Each layer (denoted by the title of the plot) contains the number of hidden denoted by the x-axis. **(right)** Comparing the deep multiplicative operation with the base cells used in the main text.

that can't be overcome by larger encoding networks. We also provide an experiment in the Directional TMaze for a **deep multiplicative** update. The deep multiplicative update uses the multiplicative update as a base cell but first passes the action through a feed forward network like the deep additive network. The results of this network were quite poor overall, likely as a result of having to learn the action encoding rather than being given it as prior information. From these results we decided to abandon the deep multiplicative extension of Zhu et al. (2017). Future work should consider how to better learn action encodings for such a network.

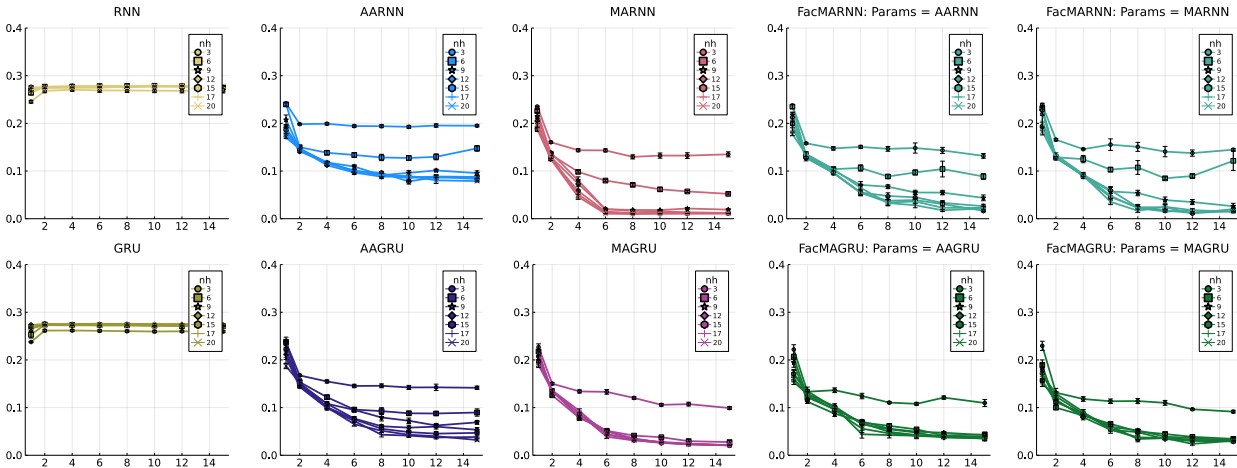

Figure 20: Truncation sensitivity curves for the Experience Replay setting in ring world. Results are RMSVE and error bars are 95% confidence, as in the main paper.)

# F   Further Empirical Details

In this section we discuss the experiments from the paper in greater detail. In the following tables the common programming notation $(x : y : z)$ is used to denote an array of elements starting from $x$ increasing by $y$ until $z$. For example $(1 : 2 : 5) = [1, 3, 5]$. When an operation is performed on an array it is done element wise. For example $2^{(1:2:5)} = [2, 4, 32]$. All hyperparameters are reported in agent steps (which are the same as environment steps for all domains).

## F.1   Ring World

Table 19 gives the hyperparameters used in the ringworld experiments. We also provide full sensitivity curves over truncation for all cell types and hidden state sizes tested in Figure 20.

## F.2   TMaze

In Figure 21 we provide details of the experience replay of the Bakker's TMaze experiments.

## F.3   DirectionalTMaze

The experiments presented in the paper used an experience replay buffer size of 10000 for all the cells. We also ran experiments using an experience replay size of 20000 with similar conclusions. These results (and the associated parameters) can be found in Figure 22

## F.4   Image Directional TMaze

We detail all hyperparameter settings, and give results for different network sizes and truncation values in Figure 23.

## F.5   Lunar Lander

We provide all hyperparameter settings (Figure 24) and further results (Figures 25 and 26)

| Parameter | Value |
|---|---|
| Steps | 300,000 steps |
| Optimizer | RMSprop |
| RMSProp $\eta$ | $0.1 \times 1.6^{(-16:3:-2)}$ |
| RMSprop $\rho$ | 0.9 |
| Buffer Size | 1000 |
| Buffer Warmup | 1000 |
| Batch Size | 4 |
| Update freq | 4 steps |
| Target Network Freq | 1000 steps |
| Independent Runs | 50 |

Figure 19: Ring World Hyperparameters

| Parameter | Value |
|---|---|
| Steps | 300,000 steps |
| Optimizer | RMSprop |
| RMSProp RNN: $\eta$ | $0.01 \times (2.0^{(-11:2:-2)})$ |
| RMSProp GRU: $\eta$ | $0.01 \times (2.0^{(-11:2:-6)})$ |
| RMSprop $\rho$ | 0.99 |
| Discount $\gamma$ | 0.99 |
| Truncation $\tau$ | 12 |
| Buffer Size | 10000 |
| Batch Size | 8 |
| Update freq | 4 steps |
| Target Network Freq | 1000 steps |
| Independent Runs | 50 |

| Cell | RMSprop Learning Rate | Hidden State Size | Number of Model Parameters |
|---|---|---|---|
| GRU | 0.005 | 6 | 214 |
| AAGRU | 0.0003125 | 6 | 286 |
| MAGRU | 0.0003125 | 6 | 754 |
| FacGRU | 0.0003125 | 6, $M = 21$ | 757 |
| DAGRU | 0.00125 | 6, $a = 4$ | 306 |
| RNN | 7.8125e-5 | 20 | 584 |
| AARNN | 7.8125e-5 | 20 | 664 |
| MARNN | 7.8125e-5 | 20 | 2024 |
| FacRNN | 0.0003125 | 20, $M = 40$ | 2064 |
| DARNN | 7.8125e-5 | 20, $a = 4$ | 684 |

Figure 21: TMaze Experience Replay experiments: **(top left)** The hyperparameters used across all cells **(bottom)** The cell specific hyperparameters **(top right)** Percent success over the final 10% of episodes. Same as Figure 9

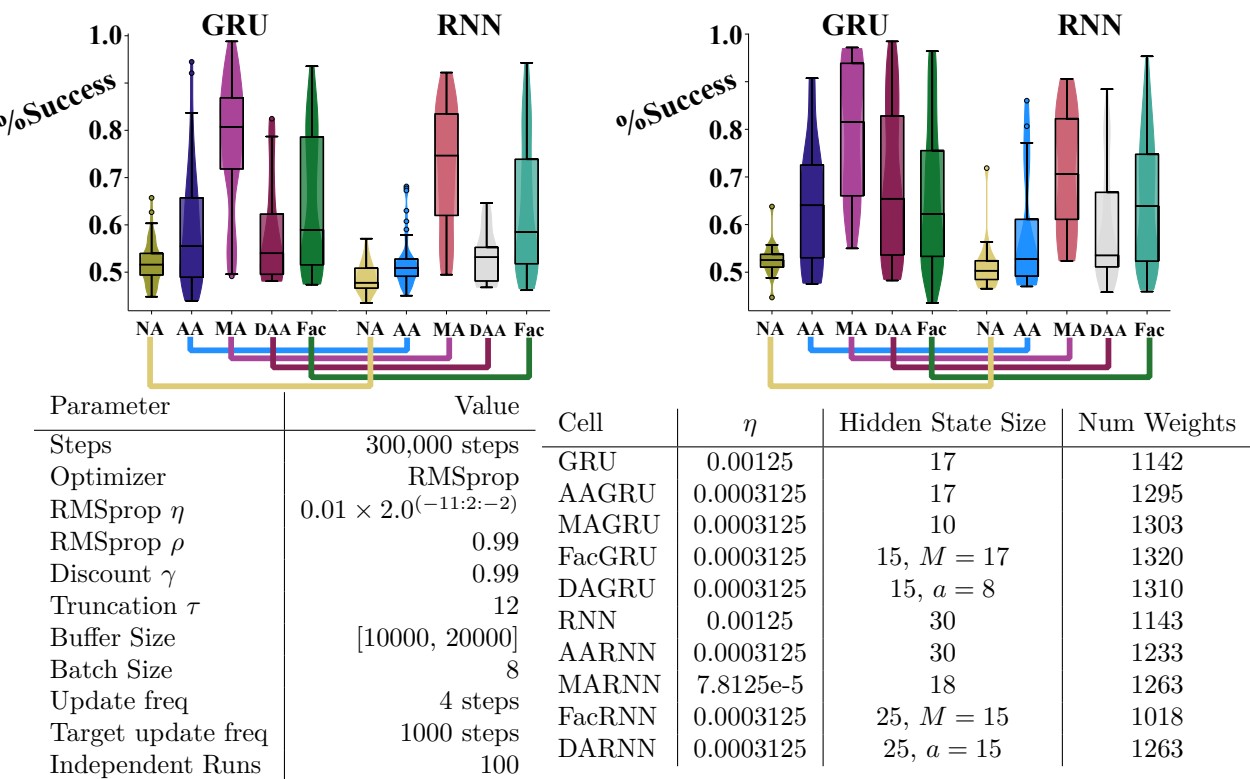

| Parameter | Value |
|---|---|
| Steps | 300,000 steps |
| Optimizer | RMSprop |
| RMSprop $\eta$ | $0.01 \times 2.0^{(-11:2:-2)}$ |
| RMSprop $\rho$ | 0.99 |
| Discount $\gamma$ | 0.99 |
| Truncation $\tau$ | 12 |
| Buffer Size | [10000, 20000] |
| Batch Size | 8 |
| Update freq | 4 steps |
| Target update freq | 1000 steps |
| Independent Runs | 100 |

| Cell | $\eta$ | Hidden State Size | Num Weights |
|---|---|---|---|
| GRU | 0.00125 | 17 | 1142 |
| AAGRU | 0.0003125 | 17 | 1295 |
| MAGRU | 0.0003125 | 10 | 1303 |
| FacGRU | 0.0003125 | $15, M = 17$ | 1320 |
| DAGRU | 0.0003125 | $15, a = 8$ | 1310 |
| RNN | 0.00125 | 30 | 1143 |
| AARNN | 0.0003125 | 30 | 1233 |
| MARNN | 7.8125e-5 | 18 | 1263 |
| FacRNN | 0.0003125 | $25, M = 15$ | 1018 |
| DARNN | 0.0003125 | $25, a = 15$ | 1263 |

Figure 22: Directional TMaze Experience Replay results: **(top right)** Percent success over the final 10% of episodes for buffer size of 10000 **(bottom right)** for buffer size of 20000. Learning rates chosen from best final performance on the final 10% of episodes for 20 runs. Results for buffer size of 10000 are over 100 independent runs, with buffer size of 20000 in appendix only over the 20 seeds used for the sweep. **(bottom left)** The hyperparameters used across all cells. **(bottom right)** The cell specific hyperparameters.

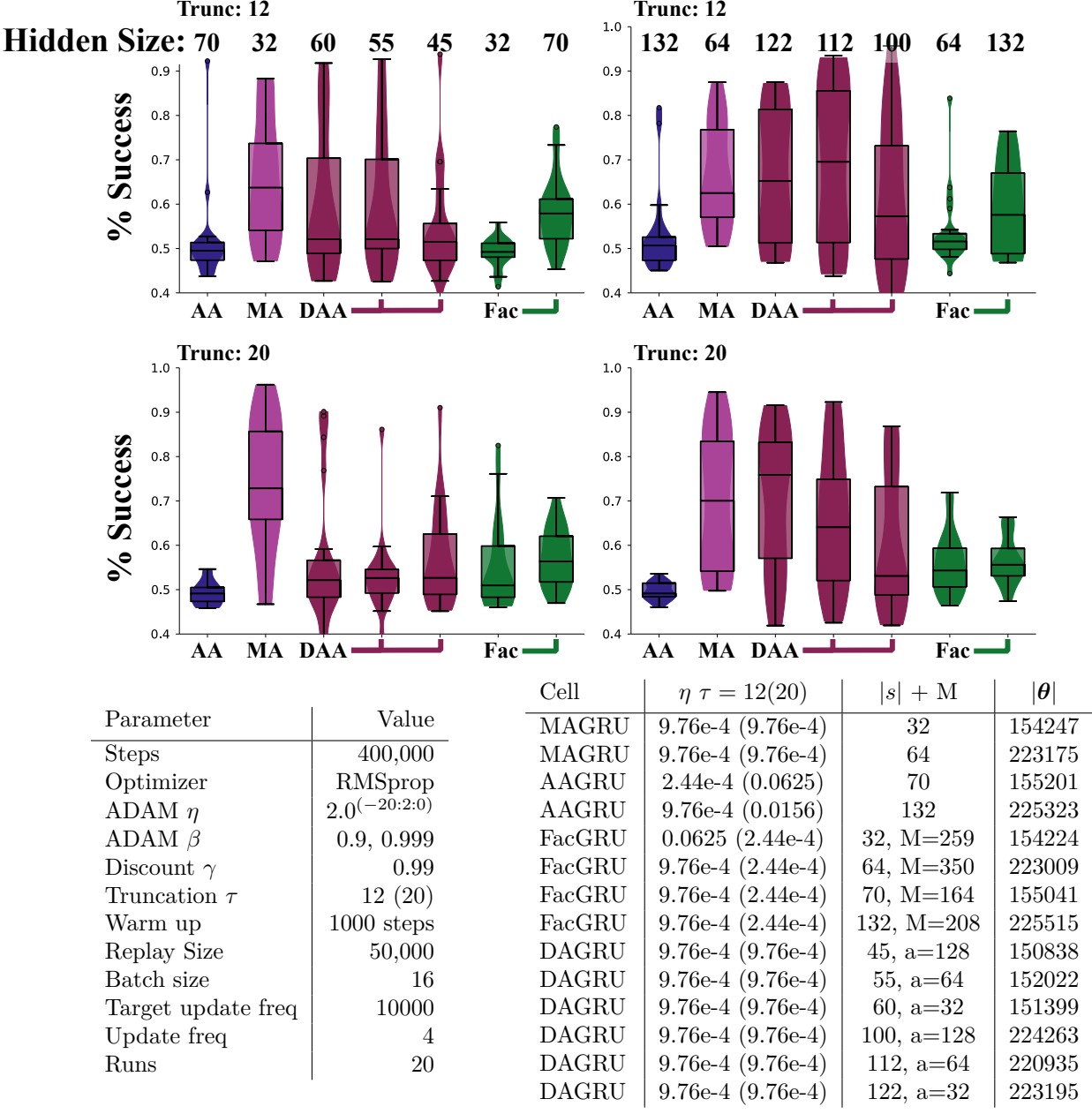

Figure 23: Image Directional TMaze: **(top)** Percent success over final 10% of episodes for the image tmaze for $\tau = 12$ and $\tau = 20$ (labeled). See labels for size of hidden state with left being small networks, and right being larger. **(bottom left)** The hyperparameters used across all cells in Image Directional TMaze **(bottom right)** The cell specific hyperparameters.

| Parameter | Value |
|---|---|
| Steps | 4,000,000 steps |
| Steps before learning starts | 1000 steps |
| Optimizer | RMSprop |
| RMSprop $\eta$ | $0.1 \times 1.6^{(-20:2:-6)}$ |
| RMSprop $\rho$ | 0.99 |
| Discount $\gamma$ | 0.99 |
| Truncation $\tau$ | 16 |
| Replay Size | 100,000 |
| Batch size | 32 |
| Target update freq | 1000 steps |
| Update frequency | 8 steps |
| Hidden state learnable | True |
| Independent Runs | 20 |

| Cell | RMSprop Learning Rate | Hidden State Size (Factors/Action Encoding) | Number of Model Parameters |
|---|---|---|---|
| GRU | 0.0003553 | 154 | 156,414 |
| AAGRU | 0.0001387 | 152 | 155,004 |
| MAGRU | 0.0003553 | 64 | 153,732 |
| FacGRU | 0.0001387 | 152 (M=170) | 152,668 |
| FacGRU | 0.0003553 | 100 (M=265) | 153,808 |
| FacGRU | 0.0003553 | 64 (M=380) | 153,716 |
| DAAGRU | 0.000138778 | 152 (a=64) | 182,684 |

Figure 24: Lunar Lander experimental details: **(top left)** The hyperparameters used across all cells in Lunar Lander **(bottom)** The cell specific hyperparameters **(top right)** Average final reward over all episodes (same as figure 12)

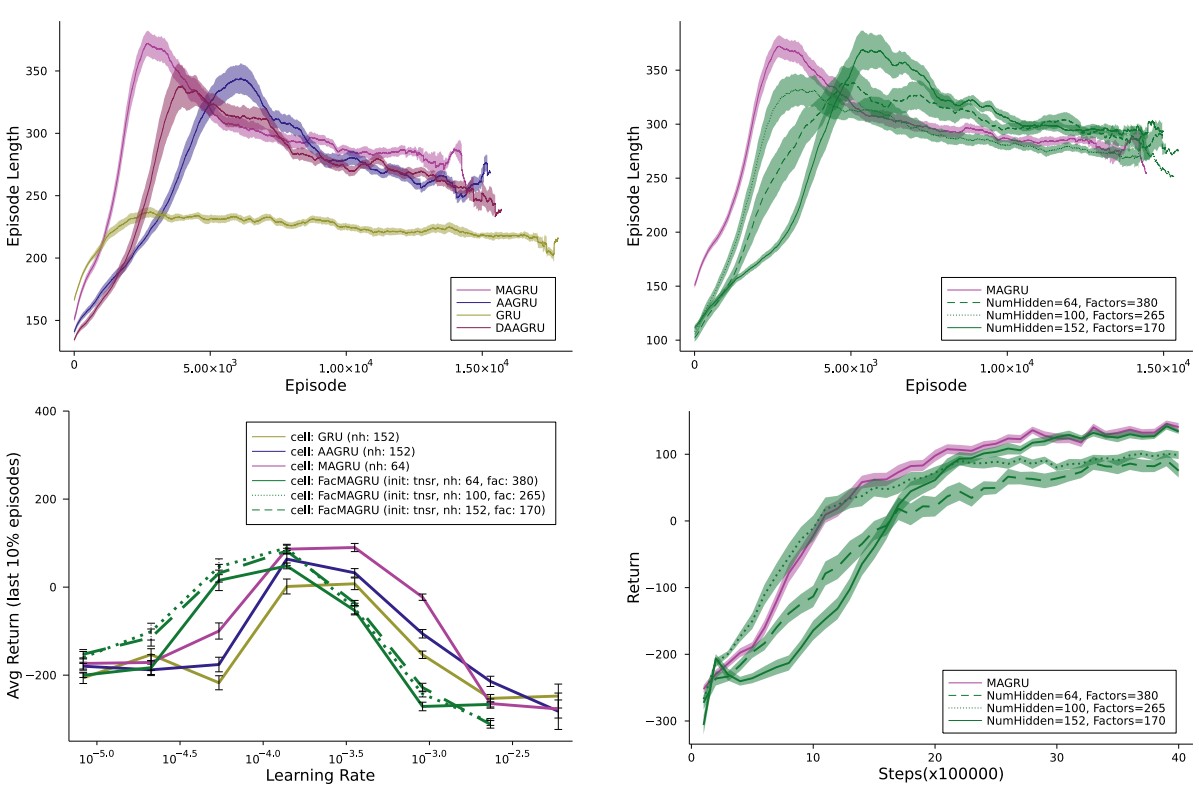

Figure 25: Lunar Lander further results: **(top left)** Average final reward over the final 10% of episodes for 20 runs **(top middle)** Total steps per episode for non-factored cells for 20 runs **(top right)** Total steps per episode for factored cells for 20 runs **(bottom left)** learning rate sensitivity curves for 10 runs **(bottom middle)** Learning curves per episode for non-factored cells over total reward for 20 runs **(bottom right)** Learning curves per episode for factored cells over total reward for 20 runs

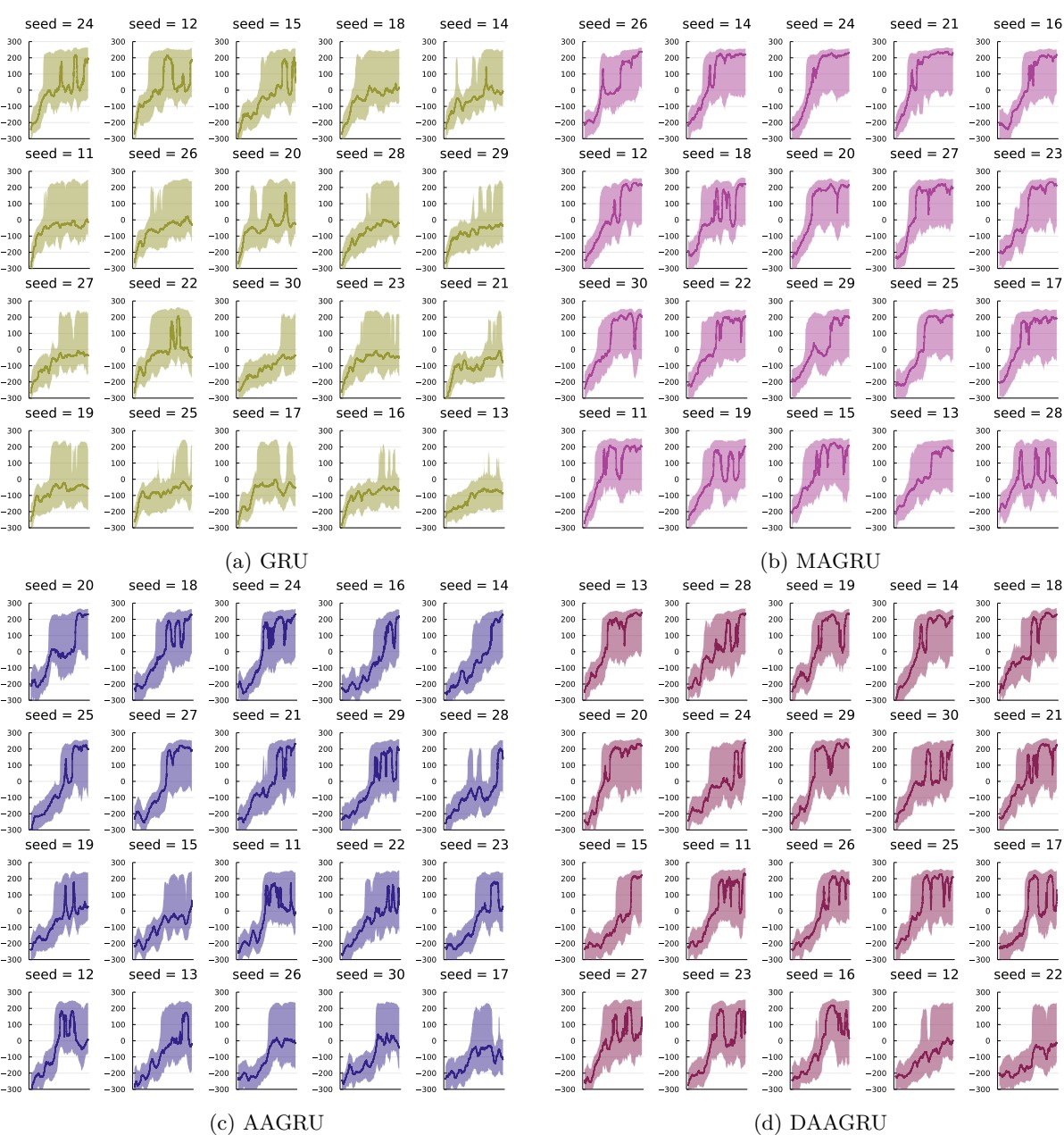

Figure 26: Individual learning curves. Line is the median over 1000 episodes, with the shaded region as the 1st and 3rd quantile over the same window.

