# OpenReview forum: "Investigating Action Encodings in Recurrent Neural Networks in Reinforcement Learning"
_TMLR — Accepted by TMLR_

### Review · Reviewer_7Jc4 · 2022-09-11

**Summary Of Contributions:**

The paper empirically studies a particular effect of architecture choices in RL training. More precisely, the question is how action should be embedded into the RNN-type networks in the POMDP setup.

In short, it advocates for using multiplicative solutions, as they provide faster and more stable training.


**Requested Changes:**

See above

**Strengths And Weaknesses:**

I have mixed feelings about this paper. In a nutshell, I like a lot the analytical approach taken assumed by the paper, however, in my view, at the current stage the paper may fall short of providing decisive evidence.

Specific comments:
- (again) I consider the approach of taking one axis and studying it in detail valuable and contrasting positively with many sota-obsessed papers
- Overall, the quality of the text is very good; a nitpick suggestion for the intro is to defer some of the discussion to related work (especially the third paragraph).
- I found it quite frustrating that the environments are not formally described. I was unsure about my understanding of the Ring World, until I checked the Internet. Even if these are generally known, it would be good to have the proper definition, even if only in the appendix.
- The environments are my major concerns. I am unsure if the conclusions in the illustrative domains, like Ring World, will transfer to more complex settings. The Lunar Lander experiment indicates that this might be true, it is hard, however, to call it exhaustive.
- Further, the learning curves in Figure 10, hint at some interesting phenomena. Some of the trainings are faster at the beginning, AA catches up at the end. It would be worth confirming whether these are merely random or general observations.
- I find Figure 5 uninformative, it is not possible to compare these curves visually; the reader needs to believe the conclusions in the text. The box plots (e.g. Fig 7) are much easier to digest, but even a simple table would do.
- Overall, I enjoy and consider valuable deeper analysis presented in the papers.


I also have a tangential question if the authors have any thoughts on the topic of 'casual confusion', see e.g. [1], which might develop in memory-augmented architectures.

[1] Causal confusion in imitation learning.

---

> ### Author Response · Authors · 2022-10-08
> **Response**
>
> Thank you for the constructive feedback! Below, we will address your specific comments in the order you have above (numbering our own). Overall, we found the suggestions to be mostly about clarity, which we have addressed through several edits to the paper (noted in red in the text).
>
> 1. We agree, focusing on a single axis and doing a detailed analysis is important to drive progress.
> 2. We believe the third paragraph adds context to the amount of work done in the supervised learning setting on constructing recurrent state update functions. We think a section explicitly titled related work is not needed because  the work we present can be seen as both an empirical study and survey of related work. Connections to prior work are discussed in the Introduction, Section 3, and throughout Section 4. In addition, we even provided a broader discussion in Appendix A, B, and C for work that is ancillary to this specific choice.  All of the cell updates we focus on have been proposed before (with some minor exceptions), and our work is a detailed account of comparing these disparate update functions.
> 3. We improved the  environment descriptions as requested. We addressed this in the main text, and added a figure for both the ring world and tmaze environments to give the readers even more context.
> 4. In this paper, we set out to test the hypothesis implicitly discussed in (Schlegel 2021), which is “The multiplicative state update performs better in the reinforcement learning setting”. We were careful not to claim the empirical evaluations done in these settings as exhaustive, as an exhaustive set of experiments would be impossible. In the domains chosen, we gain valuable insight into when one cell might be preferable to another and then test some trends using larger networks. We believe the empirical evaluations in this paper, and the conclusions we draw, are still worthwhile contributions on their own.
> 5. We agree this would be interesting to know. We report individual learning curves in the appendix (Figure 27). These results suggest that the additive doesn’t always catch up to the multiplicative in terms of median performance (albeit different than average performance). The multiplicative cell and the deep additive cells are more stable than the additive cell.
> 6. We believe these curves are informative, and give insight to the robustness of the additive and multiplicative cells over different seeds.
> 7. The work on causal confusion looks interesting! We have not thought about causal confusion, but will now be looking into this work.

---

### Review · Reviewer_hufT · 2022-09-25

**Summary Of Contributions:**

State construction is a fundamental problem in RL as the agent's behavior and predictions are based on these state features. Under the POMDP formulation, and more generally under function approximation in the standard MDP setting, high dimensional observations must be processed by the agent in order to make accurate decisions. A promising way to construct these state features is to leverage recurrent neural networks (RNNs) in order to encapsulate the agent's history and better situate itself within the world. Many design choices can be made when considering RNNs, in particular the authors study how incorporate action information, where the action correspond to the one taken by the agent in the previous state. There are essentially two families of methods: one using additive operations and another using multiplicative operations. The authors study these two families, and some variants/approximations of them in the prediction and control settings in smaller environments and larger ones. The results point towards multiplicative methods being more efficient across most settings.

**Broader Impact Concerns:**

No concerns.

**Requested Changes:**

Algorithm choices: It is not clear why the authors decided to use algorithms that are not guaranteed to converge with function approximation when performing their rigorous analysis. The algorithms I could understand that were being used is off-policy semi gradient TD and Q-Learning, which both are unstable due to the deadly triad (function approximation, off-policy and bootstrapping). Given that most environments seem to use function approximation, this really puts a doubt in terms of what we can learn from these experiments, as the usefulness of state construction is mixed with the unstability of the learning algorithm. There exists many algorithms that could be used instead of the two presented here, and for this reason I see this as a major issue. Moreover, why isn't the epsilon in Q-Learning decreased over time as is standard?

The empirical setting itself is not very clear. The authors assume that the reader will be familiar with the environments proposed, yet this shouldn't be taken for granted. For example, a nice description of Ring World or TMaze could be presented instead of Figure 5 or Figure 4 which honestly are really hard to get something meaningful out of them. In the TMaze and Ring World it is not clear what are the observations.  If the observations are Markov (i.e. not a POMDP), what is the exact role that the RNNs are achieving?

On another important point, the authors mention that additive methods and multiplicative methods have different numbers of parameters for the same number of hidden states. I would really appreciate a clear presentation of the number of parameters used by each of the methods for the presented experiments. An important question that lies within this is that perhaps a performance improvement is mostly due to the a larger network. It is also not clear what is the difference between TMaze and DirectionalTMaze, is it simply that the agent can now rotate?

There is also a proposition of combining additive and multiplicative methods, yet it is not clear why this is a proposed method. It seems like multiplicative methods are almost always better, so what can we expect to gain from this?

Results on pixel observation and sensor observations are a good contribution. In the Image DirectionalTmaze it seems like MA is slightly better, but it also suprises me that the AA agent is not able to learn. Could the authors comment if this agent eventually learns? Also, what is the motivation behind the observations given to the agent? I wonder how realistic this scenario is, for example a foraging robot could reasonably have access to a camera which would give it much richer information than the pixels described in the paper. In the Lunar Lander experiments, the authors not that there is no statistical difference between some algorithms. However, look at the plot it seems like there also isn't one when comparing MA to the rest in terms of final performance. Could the authors comment on their choice of analysis?

In general the work is lacking clarity. A related works section would help a lot in terms of situating the current contribution to past attempts. Some baselines are not explained, that is the DA baseline (or at least I couldn't find an explanation). The presentation of the multiplicative method could be made more accessible. It is required that we look up what n-mode products are, and honestly I have never come across this term up until now. What are the lambdas in the section 4.3?

There are various typos across the paper that I do not hold against the authors:
environment  section 2
significantly section 4.2
effect -> affect in the question of section 5, as well a various places later on
In this domain, we set the goal is to section 5.1
similarly section 5.1

**Strengths And Weaknesses:**

Strengths:

- Rigorous empirical evaluation, where many seeds are used and error bars are clearly identified
- Inclusion of various baselines, such as different variants for each family of methods (additive and multiplicative)
- Good effort in presenting results that pertain to different kind of environments, going from simpler domains to pixel observations

Weaknesses:

- The empirical evaluation relies on the off-policy setting, yet the algorithms used for all the experiments are not convergent or stable with function approximation.
- The empirical setting is not very clear, much information about the environments is missing. Similarly information about the number of parameters used by each method is not clearly presented.
- Related work is not complete and not well presented. It is not clear if previous has investigated a setting very similar to what is presented and what were their conclusions.

---

> ### Author Response · Authors · 2022-10-08
> **Author Response**
>
> We thank the reviewer for a constructive review of the paper. We have made several edits to the paper (which are marked in red), which address many concerns drawn by the reviewer.
>
> First we want to address one of the weaknesses raised by the reviewer directly. Specifically, “ It is not clear if previous [work] has investigated a setting very similar to what is presented and what were their conclusions.” This paper is centered on prior work, and all the main architectures we present are indeed used in prior pieces of literature (often in isolation). One main contribution of this paper is a rigorous comparison of these architectures on several illustrative domains to gain insight into which architectures are preferred and when. It so happens that the multiplicative update is the best performing, but this information was only hinted at in prior work (i.e. Schlegel, 2021). We have also provided several references for each architecture, properly placing the different approaches in the surrounding literature. We believe that separating out the related work into a separate section would be detrimental to the overall story.
>
> Below we respond to each point in the order they are listed.
>
> 1.  We chose off-policy TD(0) and Q-Learning as these are standard algorithms used for off-policy prediction and control with state-action value functions. While we agree these algorithms face the deadly triad in these settings, they are widely used in the deep reinforcement learning literature. We would also like to point out that the main motivation for submitting to TMLR is our focus on answering empirical questions rigorously and making claims in the context of the empirical evaluation. We believe this paper does that, and provide insight to the larger community who do use off-policy TD(0) and Q-Learning.
>
>       - While we stand by our choice in algorithms and don’t believe the paper needs any more empirical results to be a worthwhile contribution, we are open for suggestions on algorithms that fit the continual reinforcement learning problem setting and are convergent using recurrent neural networks. Monte carlo methods do not fit in this context. Gradient TD algorithms often are difficult to implement in ANNs (requiring second order information through the hessian) and thus aren’t widely used or studied in the deep learning context.
>
> 2. We have addressed the concern on clarity in the edits to the paper. All the environments are partially-observable.
> 3. All baselines are controlled to have approximately the same number of free parameters. In the original paper, we included detailed tables of hyper-parameters and number of free-parameters in the appendix for all experiments.
> 4. This was proposed in the context of the TMazes to think about the difference in performance between the cells. The additive combination of the two cells gives significant insight into the comparisons on Bakker’s TMaze, and provides evidence that the cells do perform similarly (unlike in the directional TMaze). We would like to remind the reviewer that our goal is not to search for SOTA, but instead expand the understanding of these different approaches and formally compare architectures informally defined in the literature.
> 5. On final performance, we agree that all the action algorithms perform similarly on average performance. When looking deeper (see figure 27 in the appendix), the multiplicative seems to be more robust over random initializations as compared to the additive. The deep additive also is much more stable, but still not as much as the multiplicative. While the final performance of the additive, deep additive, and multiplicative are similar, faster learning still makes the multiplicative more desirable.
> 6. We have clarified the reviewers issues on clarity, made the introduction to the Deep Additive architecture more clear (see section 4.2 where Deep Additive is now bolded). We added to appendix D more on n-mode products and tensors, but point to (Oh et al, 2015) and (Downey, 2017) for more examples of this notation.
>
> We thank the reviewer for the minor edits, and went through and fixed these.

---

### Review · Reviewer_LrX8 · 2022-09-30

**Summary Of Contributions:**

This paper provides a wide range of experiments supporting the observations of Schlegel et al. (2021) on how to incorporate action into the update of recurrent networks for reinforcement learning.

**Broader Impact Concerns:**

The paper does not include a broader impact statement, but I have no major concerns on its absence in this instance.

**Requested Changes:**

I believe there are valuable insights amongst the experimental results gathered in this paper, but the current write up does not do them justice. I strongly encourage the authors to (1) consider a major revision of the structure of the paper to include results from the appendix into one clear long format paper; and (2) at least address all formatting issues raised in the weaknesses section of this review.

Additionally, in discussion with the authors, I am keen to hear:

a. A more formal description of the first research question addressed by this paper.

b. A stronger case for the significance of confirming the previously published observation of Schlegel et al. (2021)

c. How this work compliments the work of Kapturowski et al. (ICLR 2019) on "RECURRENT EXPERIENCE REPLAY IN DISTRIBUTED REINFORCEMENT LEARNING."

**Strengths And Weaknesses:**

**Strengths**
1. Extensive experiments confirming the observation of Schlegel et al. (2021) of the strength of the multiplicative architecture
2. Many open research questions identified in the appendix for future work. This is a nice contribution for motivating readers to continue this line of work.


**Weaknesses**

3. The core research question introduced in section 5 is informally described as exploring the effect of these architectures on "learnability". It is of particular importance to be precise when defining the core research question of the paper.

4. Unclear conclusion on the deep action architecture. The main paper concludes "deep action versions perform marginally better" but in appendix E.5 "From these results we decided to abandon this extension of Zhu et al. (2017)'s deep action."

5. The deep action architecture is not explained in Section 4 (where the other architectures are introduced) but then referred to in Section 5.

6. The dimensions m and n in Figure 2 are not defined.

7. There are many instances of using acronyms (BPTT, RMSVE, ER, RTRL) before they are defined. Some are never defined.

8. Environments are referred to before being defined. For example, in Section 5.1 the paper states we are "revisiting the Ring World environment" but we haven't yet been introduced to the environment.

9. Figure 6 and 13 are hard to see the important details conveyed in colour due to the size of images. Both take up a whole page with significant white space surrounding the figures. The clarity could be improved by scaling up the images to use the full page.

10. Figure 7 again refers to environments not yet defined, but more significantly I think the caption has the labels for left and right the wrong way around as the results are inconsistent with the text that summarizes them in the main body of the paper.

11. The appendix is a disorganized collection of additional experiments. Some appear to add significant value but are not sufficiently described in context with the rest of the paper and, in the extreme example of the deep action architecture, appear to contradict the main paper.

12. Appendix D appears almost entirely irrelevant, with only the last paragraph in a 2 page section adding some vague new insight in-line with the topic of the paper.

13. Appendix E.1 and E.3 discuss results without referencing which figure they are referring to. Similarly, appendix E.5 refers to "The results of this network" ambiguously without clearly referring to (I assume from the following sentence) the deep action architecture.

14. Figure 12 has no legend. What are the blue, purple and pink lines?

15. Appendix E.4 criticizes running experiments in just TMaze and Ring World, but the main paper has results also in lunar lander. This seems like an unfair criticism of the main paper, and it is confusing that this result is pushed to the appendix when it is addressing a weakness the authors have identified in their own paper.

16. The formatting of Figures 16 and 17's captions is unnecessarily confusing given the white space available on the page.

---

> ### Author Response · Authors · 2022-10-08
> **Response to concerns**
>
> We thank the reviewer for a detailed and extensive review of the main text and appendix. Below we will respond to each of the points made in the weaknesses section. Overall, we have edited the paper (edits marked in red) and look forward to the reviewers thoughts on the improvements to clarity. Below we also address the reviewers suggestions to changes.
>
>
>
>
> 3. We added two new subsections detailing the research questions more specifically. We agree this adds clarity to the paper overall. Thank you for the suggestion.
> 4. There were clarity issues in Appendix E.5. The DA presented in the paper corresponds to “Deep Additive” not “Deep Action”. In appendix E.5 we are looking at the performance of the Deep Additive architecture over different action embedding network depths, and how well a “Deep Multiplicative” (DM) style architecture performs. The DM architecture was not presented in (Zhu et al. 2017), so we didn’t pursue the architecture for the reasons mentioned (i.e. learning an appropriate action embedding requires more work). We have rewritten the section to make this more clear.
> 5. The Deep Additive was explained in section 4.1 with the additive approach. We make this more clear in that section.
> 6. We make sure to define these through the main text and in the figure caption.
> 7. We have addressed this and defined all the initialisms
> 8. We have revised how we introduce the environments and added illustrations to further clarify the illustrative domains.
> 9. We agree. For Figure 6 we split the plot onto two pages (based on the different seeds). We don’t have a solution for figure 13 currently, but are open to suggestions.
> 10. Thank you for the edit, we have fixed the ordering in the paper.
> 11. We have addressed the issues with appendix E.5. Section E was exploration we did as a part of this paper, but the results weren’t interesting or clear conclusions couldn’t be drawn. While we don’t think these add to the main contributions of the paper, we think they are useful for those following this line of research.
> 12. Appendix D was to help explain the notation used in the Multiplicative update and the Factored update. We clarified this at the beginning of the section.
> 13. We have fixed this.
> 14. We have addressed this as well.
> 15. We agree, we are addressing a weakness with the main paper. But the insights drawn from this environment are limited and confirm the trend of conclusions drawn in the main paper. While worthwhile reporting, we think the experiment doesn’t add any significant insight over those presented in the main paper.
> 16. We agree with the reviewer and fix this in the edited paper.
>
> Requested changes:
>
> We have addressed the formatting concerns raised by the reviewer. Thank you for helping us make the paper clearer and more impactful. We don’t believe all the results in the appendix bring more insight into the conversation, and many (especially in Appendix E) are explorations that often lead to a dead-end. We reported these results in good faith as the experiments were run, the conclusions are not always straightforward/insightful, and a path forward on these topics is not clear.
>
> While we think most of the experiments should stay in the appendix, we are open to suggestions for adding more of the results to the main paper. We look forward to having this conversation with the reviewer.
>
> Additional comments:
>
> (a) We have added a more formal description of the research questions, and look for feedback on the clarity brought from these descriptions.
>
> (b) The multiplicative update is not only used by Schlegel (see (Goudreau 1994), (Rafols 2006), (Sutskever, 2011), (Downey, 2017), and implicitly (Oh 2015)), but (Schlegel 2021) does notice this choice and present the use of the multiplicative update in a vanilla RNN for reinforcement learning first (as far as we are aware).
>
> (c) This is a highly relevant piece of literature for section C we missed. Generally, the architectural choices on how to use the experience replay buffer (and update the hidden state in the buffer) should not affect the conclusions we draw here. Our strategy using the ER buffer is somewhat different from Kapturowski, but we aren’t in the distributed setting and are more focused on the single agent single environment sequential setting. We also don’t believe the conclusions drawn will be heavily impacted by this decision (as discussed in the appendix). We report this in the main text now as well.

---

### Author Response · Authors · 2022-10-08
**Revision and Reviews**

Thank you all for your insightful and constructive comments. We have added a new revision to the paper. The changes from the previous paper are marked in red. We will address other concerns raised by the reviewers individually. We look forward to further discussion,

---

### Decision · Action_Editors · 2022-12-14

**Recommendation:** Accept as is

**Comment:**

Being somewhat of an empirical study paper, the experimental method is particularly important, and reviewer hufT raises the valid concern that the choice of evaluating these methods with Q-learning and TD(0) may not have been optimal. However, I believe that these are reasonable choices, and improving this would require significant additional work, beyond the scope of a minor revision.

Clarity has been a problem with this paper through the review stage. However, through many revisions addressing the quite extensive feedback from reviewers, it has now improved sufficiently.

As such, I am recommending "accept as is". If possible, and if the authors so choose, the authors may additionally acknowledge the efforts of the anonymous reviewers towards improving the manuscript.

**Audience:**

The questions and findings in this paper are likely to be of wide interest to TMLR, and to RL researchers in particular.

**Claims And Evidence:**

Current RL approaches for POMDPs often use hidden state of an RNN processing the observation-action history as the state representation, but the nitty gritty is not very well-studied. This paper studies the hypothesis that how precisely past *actions* are integrated into the state history is particularly important, and studies various possible choices among "additive" and "multiplicative" approaches to doing so, identifying a type of multiplicative architecture (first suggested in Schlegel et al 2021) as a widely performant design choice, and the conditions where this choice matters most.

---

> ### Author Response · Authors · 2022-12-20
> **Thank you**
>
> Thank you for the comments! We've added acknowledgements to the reviewers and all the necessary details for the camera ready version (as well as some more minor edits). We've also included links to a short video and code repo!
>
> Thanks again for a great review process!